# Chromatin activity of IκBα mediates the exit from naïve pluripotency

Luis G Palma[1,2,3†], Daniel Alvarez-Villanueva[1,4†], Maria Maqueda[1,2,3], Mercedes Barrero[5], Arnau Iglesias[1,2,3], Joan Bertran[6], Damiana Alvarez[2], Carlos A Garcia-Prieto[2], Cecilia Ballare[5], Virginia Rodriguez-Cortez[2], Clara Bueno[2], August Vidal[4], Alberto Villanueva[7], Pablo Menendez[2,3,8,9,10], Gregoire Stik[2], Luciano Di Croce[5,10,11], Bernhard Payer[5,11], Manel Esteller[2,3,10,12], Lluis Espinosa[1,3]*, Anna Bigas[1,2,3]*

[1]Program in Cancer Research. Hospital del Mar Research Institute, Barcelona, Spain; [2]Josep Carreras Leukemia Research Institute, Barcelona, Spain; [3]Centro de Investigación Biomédica en Red de Cáncer (CIBERONC), Madrid, Spain; [4]Institut Investigació Biomèdica de Bellvitge (IDIBELL), L'Hospitalet de Llobrega, Barcelona, Spain; [5]Centre for Genomic Regulationlation(CRG), The Barcelona Institute of Science and Technology, Barcelona, Spain; [6]Universitat de Vic - Universitat Central de Catalunya, Vic, Spain; [7]Catalan Institute of Oncology (ICO/IDIBELL), L'Hospitalet de Llobregat, Barcelona, Spain; [8]Spanish Network for Advanced Therapies (RICORS-TERAV). Carlos III Health Institute (ISCIII), Madrid, Spain; [9]Department of Biomedicine. University of Barcelona, Barcelona, Spain; [10]Institucio Catalana de Recerca i Estudis Avançats(ICREA), Barcelona, Spain; [11]Universitat Pompeu Fabra, Barcelona, Spain; [12]Physiological Sciences Department, School of Medicine and Health Sciences, University of Barcelona (UB), Barcelona, Barcelona, Spain

*For correspondence:
lespinosa@researchmar.net (LE);
abigas@researchmar.net (AB)

†These authors contributed equally to this work

## eLife Assessment

This **important** study describes a non-canonical role for IκBα in regulating mouse embryonic stem cell pluripotency and differentiation, independent of the classical NF-κB pathway. The conclusions are **convincingly** supported through orthogonal approaches and separation of function mutants. The findings add new insight into pluripotency regulation in mouse cells.

**Abstract** Maintenance of pluripotency is a multifactorial process in which NF-κB is a negative regulator. Our previous work identified a chromatin role for IκBα, the master regulator of NF-κB signaling, that is critical for the proper regulation of various tissue stem cells. Here, we found that IκBα accumulates specifically in the chromatin fraction of mouse pluripotent stem cells. IκBα depletion does not affect NF-kB-dependent transcription, but causes a profound epigenetic rewiring in pluripotent stem cells, including alterations in H3K27me3, a histone mark catalyzed by Polycomb repression complex 2. Chromatin changes induced by IκBα depletion affect a subset of pluripotency genes and are associated with altered gene transcription. At the cellular level, IκBα-deficient embryonic stem cells are arrested in a naive pluripotency state when cultured in serum/LIF conditions and fail to exit pluripotency under differentiation conditions. By constructing separation-of-function mutants, we show that the effects of IκBα in regulating stem cell pluripotency are NF-κB-independent, but mainly rely on its chromatin-related function. Taken together, our results reveal a novel mechanism by which IκBα participates in the regulation of the pluripotent state of mouse embryonic stem cells and shed light on the interplay between inflammatory signals and the regulation of pluripotency.

## Introduction

Embryonic Stem Cells (ESCs) pluripotency is characterized by the capacity to generate all somatic and germline lineages both in vitro and in vivo, depending on the culture conditions. When cultured in serum plus Leukemia Inhibitory Factor (LIF) (Serum/LIF), murine ESCs (mESCs) include cells interconverting to different metastable states (*Festuccia et al., 2013*) that either resemble preimplantation blastocyst (naïve pluripotency) or post-implantation (primed pluripotency) embryonic stages (*Graf and Stadtfeld, 2008*; *Martinez Arias and Brickman, 2011*; *Pera and Rossant, 2021*; *Wang and Wu, 2022*). The heterogeneity of mESCs in Serum/LIF is reduced by culturing them in the presence of GSK3β and MEK1/2 inhibitors (CHIR99021 and PD0325901, respectively) along with LIF (2i/LIF). Under these conditions, mESCs enter a ground state of naïve pluripotency characterized by self-renewal activity while suppressing any pro-differentiating signals (*Ying et al., 2008*; *Marks et al., 2012*). This pluripotent state closely resembles the inner cell mass of the E4.0 mouse preimplantation blastocyst (*Boroviak et al., 2015*). After implantation, pluripotent capability of the mouse embryo (E5.5) becomes restricted to Epiblast Stem Cells (EpiSCs), which represent a primed pluripotent state, characterized by coexpression of pluripotency and lineage specification markers (*Brons et al., 2007*; *Tesar et al., 2007*). EpiSCs can give rise to cells from the three germ layers (endoderm, mesoderm, and ectoderm), albeit their contribution to chimeras or to germ cells is greatly compromised (*Pera and Rossant, 2021*; *Weinberger et al., 2016*). EpiSCs can be derived in vitro from naïve pluripotent cells when cultured in a medium supplemented with Nodal and fibroblast growth factor (FGF) (*Guo et al., 2009*). Different epigenetic features distinguish naïve from primed pluripotent states (*Takahashi et al., 2018*), such as changes in the X chromosome inactivation in female cells (*Guo et al., 2009*; *Bao et al., 2009*), histone post-translational modifications (*Tesar et al., 2007*), or a global DNA hypomethylation profile observed in naïve pluripotent stem cells (*Habibi et al., 2013*; *Leitch et al., 2013*; *Singer et al., 2014*). These epigenetic changes are a prerequisite for the subsequent activation of gene circuits associated with pluripotency exit and germ layer specification (*Brons et al., 2007*; *Tesar et al., 2007*). Crucial regulatory elements for the transition from naïve to primed pluripotency state are located in distal DNA regions or enhancers to facilitate the resolution toward the primed state (*Factor et al., 2014*; *Buecker et al., 2014*). Therefore, identifying novel players to fine-tune the equilibrium between ground and primed states is crucial for preserving both pluripotency stability and differentiation capability in mESCs.

There is now strong evidence that inflammatory signals are critical for stem cell development (*Gerondakis et al., 2006*; *Kaltschmidt et al., 2021*; *Espín-Palazón and Traver, 2016*), with NF-κB being the main effector. Different strategies to inhibit or attenuate NF-κB in ESCs lead to increase in pluripotency gene expression and impair cell differentiation (*Li et al., 2024*; *Torres and Watt, 2008*), or it facilitates reprogramming to induced pluripotent stem cells (iPSC) (*Dutta et al., 2011*). Canonical NF-κB signaling is triggered by the IKK kinase complex, which induces the phosphorylation and subsequent degradation of the NF-κB inhibitor, IκBα, leading to nuclear translocation of the NF-κB factors (e.g. p50/p65) (*Zhang et al., 2017*). However, we have previously identified a nuclear IκBα function, which is critical for tissue stem cell homeostasis and the proper differentiation of epidermal, intestinal, and hematopoietic stem cells (*Mulero et al., 2013*; *Marruecos et al., 2020*; *Marruecos et al., 2021*; *Thambyrajah et al., 2024*). IκBα chromatin function is mediated by the interaction with histone deacetylases (HDACs), core elements of the Polycomb Repressor Complex 2 (PRC2), and histone H2A and H4 (*Mulero et al., 2013*; *Marruecos et al., 2021*; *Aguilera et al., 2004*).

We have now investigated the role of IκBα in the context of pluripotent stem cells. Whereas cytoplasmic IκBα is detected in the different states and its levels decrease after differentiation, in agreement with NF-κB activity as a promoter of differentiation, we specifically detected chromatin-bound IκBα in the naïve pluripotent stem cells. Importantly, IκBα depletion in ESCs did not affect NF-κB activity but it stabilizes the naïve (2i/LIF-like) state of pluripotency under Serum/LIF conditions, displaying an epigenetic rewiring, including changes in the PRC2-dependent H3K27me3 histone mark and impacting proximal and distal regulation of pluripotent genes. Using a new Separation-Of-Function (SOF) IκBα mutants, we have now definitively demonstrated that chromatin-bound IκBα is required for naïve pluripotency exit of mESCs in an NF-κB-independent manner. Overall, we identified IκBα as a novel key player in the regulation of the exit from naïve-to-primed pluripotency and essential for the activation of gene programs involved in germ layer specification.

# Results

## IκBα and NF-κB have opposite expression dynamics in naïve pluripotent cells

Due to the prominent role of NF-κB signaling in mESCs regulation, we analyzed the expression pattern of the different NF-κB members, including the IκB inhibitors in the three defined pluripotent stages: ground state of naïve pluripotency (2i/LIF), naïve pluripotency (Serum/LIF), and primed pluripotency (Epiblast Stem Cells or EpiSCs) (*Wang and Wu, 2022*; *Figure 1A*). In agreement with NF-κB being a pro-differentiation factor, canonical NF-κB inhibitors, including IκBα (*Nfkbia*), IκBβ (*Nfkbib*), and IκBε (*Nfkbie*) were all expressed in the naïve pluripotency state (both 2i/LIF and Serum/LIF), with *Nfkbia* levels gradually reduced upon naïve pluripotency exit towards EpiSCs (*Figure 1B*). In contrast, *Rela, Nfkb1, and Nfkb2* genes, codifying for the NF-κB factors p65/RelA, p105, and p100, respectively, showed opposite expression dynamics increasing the levels in EpiSCs (*Figure 1C*). We further characterized the distribution of IκBα in the cytoplasm, nucleus and chromatin fractions of the three states of pluripotency (*Figure 1D*). Total IκBα protein levels correlated with the detected RNA levels in the three different cellular states being highest in the 2i/LIF condition (*Figure 1B*), with a large accumulation in the cytoplasm. A fraction of IκBα protein was detected in the chromatin of naïve pluripotent cells, being highest in cells cultured in Serum/LIF conditions compared to 2i/LIF and absent in the EpiSCs. (*Figure 1D*). The dynamics of IκBα RNA and protein levels from naïve pluripotent ESCs to EpiSCs was further corroborated by analysis of an additional database that includes both naïve and primed pluripotent states (*Atlasi et al., 2020*; *Figure 1E*).

Next, we studied IκBα expression in differentiated embryoid bodies (EBs) from mESCs (Serum/LIF) (see materials and methods) and analyzed the expression levels of different pluripotency, differentiation-linked, and NF-κB genes by RNA sequencing. *Nfkbia* (IκBα gene) was expressed at higher levels specifically in mESCs (Serum/LIF), whereas cells undergoing differentiation (48 hr and 96 hr) displayed a reduction in its expression levels (*Figure 1F*). The expression pattern of *Nfkbia* was similar to the pluripotency genes, whereas most of the NF- κB genes were upregulated upon differentiation, showing an analogous expression dynamics as developmental genes, as previously described (*Torres and Watt, 2008*). Previous reports have demonstrated that chromatin-bound IκBα modulates PRC2 activity in different adult stem cell models (*Mulero et al., 2013*). Notably, we observed that most of the Polycomb target genes follow a similar expression pattern of *Nfkbia* and pluripotency, with higher expression in mESCs (*Figure 1F*).

Taken together, these data show that, unlike the other NF-kB inhibitors and target genes, the expression of *Nfkbia* is high in the naive pluripotent state, with IκBα protein present in the cytoplasm and chromatin of naive pluripotent cells. Moreover, whereas canonical NF-κB subunits are primarily absent, we detected several Polycomb elements expressed in mESCs.

## IκBα is required in mESCs to exit the naïve pluripotency state

To investigate the functional implication of IκBα in naïve pluripotency, we knocked out (KO) v IκBα protein in mESCs by CRISPR-Cas9 (IκBα-KO mESCs) (*Figure 2—figure supplement 1A* and Materials and methods). Notably, IκBα-KO mESCs have an impaired ability to undergo differentiation and cannot properly switch off the pluripotency program after 216 hr (9 days) of pro-differentiation signals, as it is demonstrated by high levels of pluripotency markers OCT3/4 and NANOG (*Figure 2A* and *Figure 2—figure supplement 1C*), higher number of alkaline phosphatase cells (*Figure 2B* and *Figure 2—figure supplement 1D*) and higher percentage of SSEA- cells withinin the 216 hr EBs. These results were also supported by the detection of increased expression level of the naïve pluripotency genes *Pou5f1 (Oct3/4 gene), Gbx2, Klf2, Sox2, Zfp42 (Rex1)*, and *Nanog* in IκBα KO EBs after 216 hr of differentiation (*Figure 2—figure supplement 1F*) and lower differentiation (*Figure 2—figure supplement 1G*).

To shed light into the molecular mechanisms governing the differentiation impairment in IκBα KO EBs, we performed RNA-seq of IκBα WT and IκBα KO mESCs in the first stages of the embryoid body differentiation. Analysis of the data indicated that IκBα KO EBs cannot successfully evolve towards differentiation trajectories, as shown by principal component analysis (PCA) of IκBα-WT and IκBα-KO EBs at 48 hr and 96 hr in differentiation media (*Figure 2C*). Analysis of naïve pluripotency markers at 96 h (*Gab1, Sox2, Esrrb, Zfp42, Dppa4, Klf2, Prdm14, Klf4, Tfcp2l1*) also showed higher expression levels in IκBα KO compared to their IκBα WT counterparts (*Figure 2D* and *Figure 2—figure*

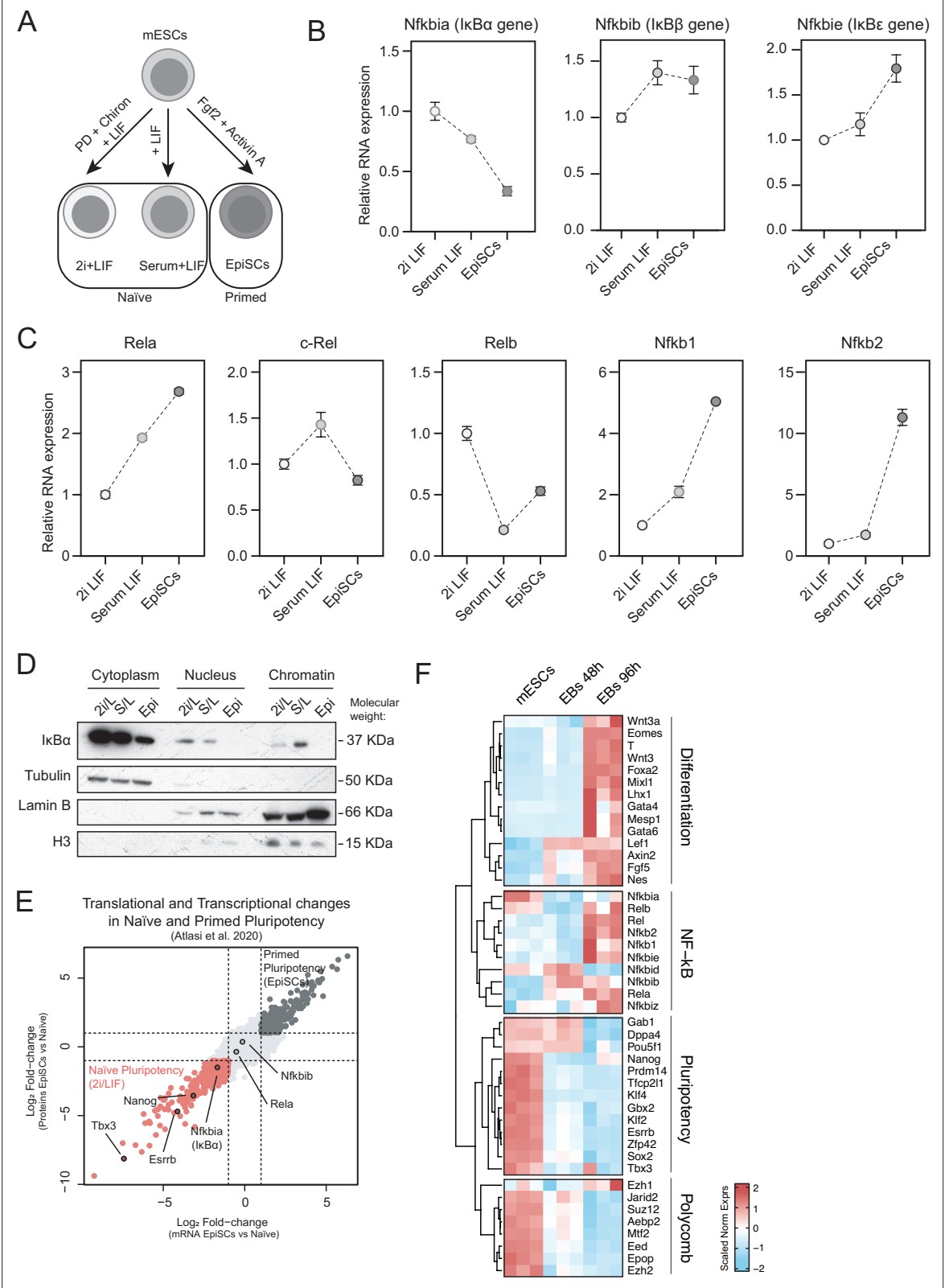

**Figure 1.** IκBα is highly expressed in pluripotent cells. (**A**) Schematic of pluripotent states in mouse embryonic stem cells (mESCs). mESCs were maintained regularly in naïve pluripotency by culturing them in serum/Leukemia Inhibitory Factor (LIF) medium, or they were polarized towards the ground state of naïve pluripotency by culturing them for two passages in LIF plus Gsk3β (CHIR99021 or Chiron) and MEK (PD0325091) inhibitors (2i/LIF). mESCs were further differentiated towards the primed pluripotent state (epiblast-like cells) by culturing mESCs in a medium containing 20 ng/mL Activin

*Figure 1 continued on next page*

*Figure 1 continued*

A and 10 ng/mL Fgf2 for 120 hr. (**B–C**) Relative mRNA levels (based on qPCR experiments) of the canonical IκB genes (*Nfkbia, Nfkbib, Nfkbie*) (**B**) or Nf-κB effector genes (*Rela, c-Rel, Relb, Nfkb1, Nfkb2*) (**C**) across the different states of pluripotency. Expression was normalized using the house-keeping gene *Tbp* relative to 2i/LIF. Data from three independent experiments. Dots indicate mean values and error bars refer to ± standard deviation (SD). (**D**) Western blot analysis of IκBα protein distribution in cytoplasm, nucleoplasm (Nucleus), or chromatin in cells at the ground-state of naïve pluripotency (2i/L), naïve pluripotency (S/L), or primed pluripotency (Epi). This experiment was independently repeated twice. (**E**) Transcriptome and proteome differences across naïve (2i/LIF) and primed (Epiblast-like cells or EpiSCs) pluripotent states. Enriched transcripts and proteins in naïve state are highlighted in red (Log$_2$ fold-change < –1). Enriched transcripts and proteins in the primed state are marked in dark gray (Log$_2$ fold-change > 1). Data from ***Atlasi et al., 2020***. (**F**). Heatmap showing the expression levels of pluripotency (*Dppa4, Pou5f1, Sox2, Prdm14, Klf4, Zfp42, Nanog*), differentiation (*Mesp1, Gata4, Gata6, Lhx1, Wnt3a, Fgf5, Nes, Axin2, Wnt3, Eomes, Foxa2, T, Mixl1, Lef1*) NF- κB (*Nfkbia, Nfkbib, Nfkbie, Nfkbiz, Rela, c-Rel, Relb*), and Polycomb (*Ezh1, Ezh2, Eed, Epop, Jarid2, Suz12, Mtf2, Aebp2*) genes from the RNAseq samples at mESCs (Serum/LIF) and differentiating cells (embryoid bodies) at 48 hr and 96 hr of IκBα-WT cells. Normalized counts based on z-score are represented.

The online version of this article includes the following source data for figure 1:

**Source data 1.** Annotated files for western blot analysis displayed in ***Figure 1D***.

**Source data 2.** Raw files for western blot displayed in ***Figure 1D***.

***supplement 1C***). Defective differentiation capacity of IκBα KO Ebs involved programs associated with specification of all three germ layers (endoderm, mesoderm, and ectoderm) (***Figure 2E***). Moreover, IκBα KO EBs were smaller in size (***Figure 2—figure supplement 1E***), in agreement with the reduced developmental potential of IκBα-depleted mESCs (***Tian et al., 2019***).

To further study the requirement of IκBα in pluripotency exit in vivo, we established teratomas by subcutaneous injection of IκBα-WT and IκBα-KO mESCs (***Figure 2F*** and Materials and methods) into NSG (NOD.Cg-Prkdc[scid] Il2rg[tm1Wjl]/SzJ) mice. IκBα-KO teratomas showed a decreased differentiation potential, as indicated by higher number of OCT4[+] cells 6 weeks after injection (***Figure 2G–H***). These observations further confirm the essential role of IκBα in the exit from the pluripotency state not only in vitro but also under in vivo differentiation signals. To understand the conditions in which IκBα is required for pluripotency exit, we differentiated IκBα-WT and IκBα-KO mESCs cultured in Serum/LIF (naïve pluripotency) towards EpiSCs (primed pluripotency) (***Figure 3A***, upper panel). IκBα-KO cells were not committed towards the primed pluripotency stage and maintained an elevated expression levels of naïve pluripotency genes (*Dppa3, Nanog, Sox2, Rex1, Klf2, Klf4, Gbx2, Tbx3*) (***Figure 3A***, bottom panel). Additionally, IκBα-KO cells retained a naïve pluripotent morphology, forming tight clusters that resemble cells cultured in 2i/LIF medium (***Figure 3—figure supplement 1A***).

mESCs cultured in serum/LIF are highly heterogeneous, comprising a mixture of cell states that resemble developmental transitions from preimplantation (naïve pluripotency) to postimplantation (primed pluripotency) embryo (***Graf and Stadtfeld, 2008***; ***Martinez Arias and Brickman, 2011***). To investigate how IκBα deficiency was affecting the exit of mESCs from naïve pluripotency, we performed a Gene Set Enrichment Analysis (GSEA) using RNA-seq data from IκBα-KO and IκBα-WT mESCs cultured in Serum/LIF against gene signatures for 2i/LIF and Serum/LIF pluripotency (***Ghimire et al., 2018***). GSEA results revealed a significant enrichment in Serum/LIF IκBα-KO mESCs transcriptome for genes specifically expressed in the naïve ground state (2i/LIF), while they negatively correlate with mESCs cultured in Serum/LIF (***Figure 3B***; ***Ghimire et al., 2018***). These results agree with the fact that IκBα-KO mESCs cultured in Serum/LIF resemble 2i/LIF morphology, with homogeneous tight clusters of cells, whereas IκBα-WT mESCs retain the colony heterogeneity (***Figure 3—figure supplement 1A and B***). Interestingly, the higher levels of ground-state-related genes *Zfp42* (*Rex1*), *Klf2* and *Tbx3* in IκBα-KO mESCs were partially obtained when IκBα-WT mESCs were cultured in 2i/LIF for two consecutive passages (***Figure 3C***).

One of the key features of the naïve pluripotency state is its global DNA hypomethylation pattern (***Habibi et al., 2013***; ***Leitch et al., 2013***). We assessed the DNA methylation status of IκBα-WT and IκBα-KO mESCs cultured in Serum/LIF conditions by analysis of 5-methyl-cytosine (5 mC) mark and using DNA methylation arrays (***Zhou et al., 2022***). We found that IκBα-KO mESCs contained much lower levels of 5 mC compared to IκBα-WT cells (***Figure 3D***). Moreover, DNA methylation arrays confirmed a global pattern of DNA hypomethylation in IκBα-KO mESCs, which was maintained in 96 hr EBs (***Figure 3E***). Interestingly, the DNA hypomethylation status occurred in a genome-wide fashion affecting all chromosomes (***Figure 3—figure supplement 1C***), in accordance with a higher pluripotent state of IκBα- KO mESCs.

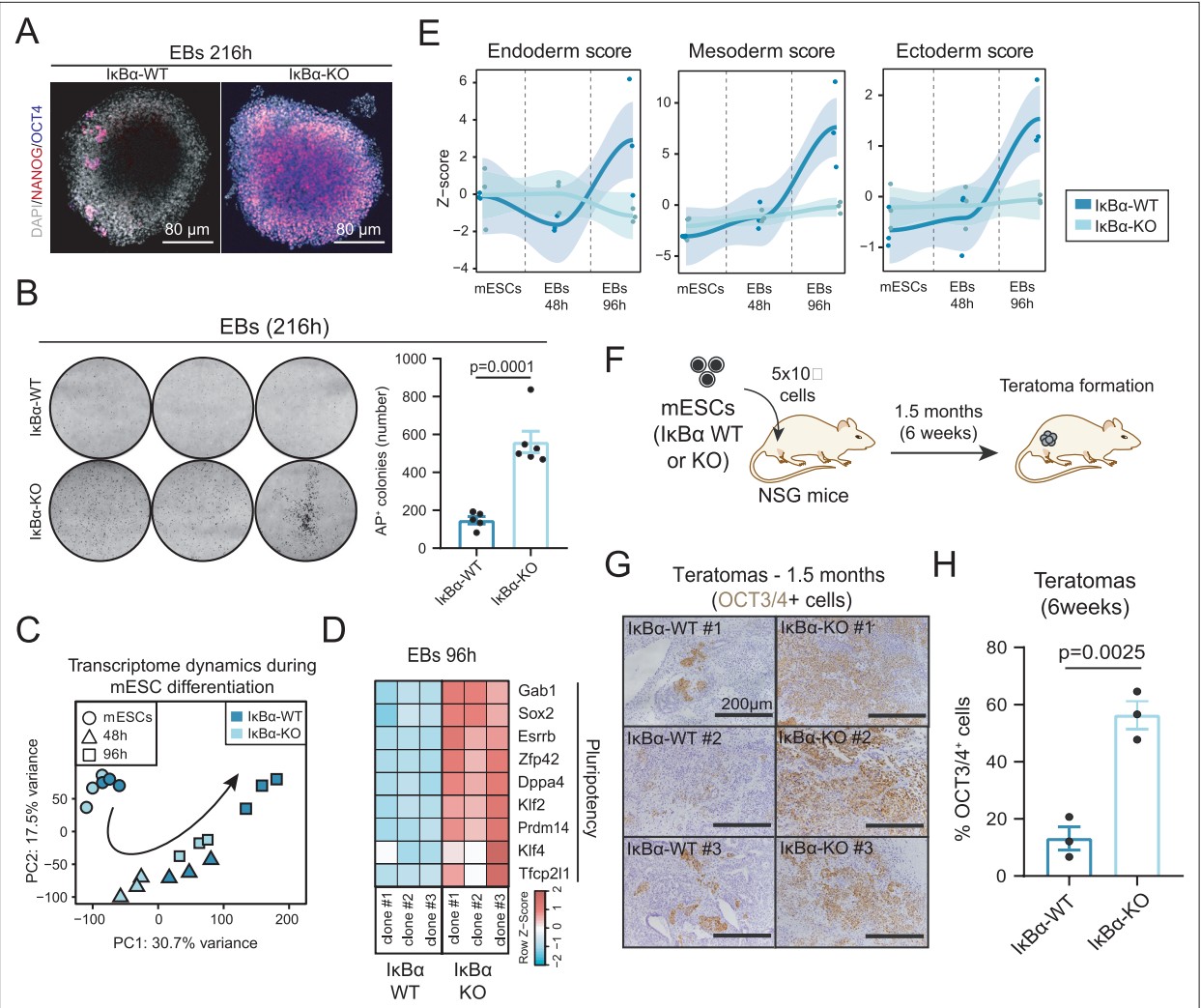

**Figure 2.** The absence of IκBα hinders the exit from the pluripotent state and prevents the activation of differentiation programs. (**A**) Representative immunofluorescence of 216 hr embryoid bodies (EBs) stained with the pluripotency markers NANOG and OCT3/4. (**B**) Quantification of total number of alkaline phosphatase (AP) colonies after 216 hr EB differentiation and additional culture of 96 hr in Serum/LIF (S/L) medium. Dots represent values from two independent experiments from three different IκBα-WT or IκBα-KO clones. Unpaired two-sided t-test applied. Representative images from alkaline phosphatase (AP) colonies are included in the upper panel. (**C**) PC1 and PC2 from Principal Component Analysis (PCA) of normalized RNA-seq data from three different time points (mESCs, 48 hr EBs, and 96 hr EBs) of IκBα-WT and IκBα-KO cells (three independent clones from each genotype). (**D**) Heatmap showing the expression levels of naïve pluripotency genes (*Gab1, Sox2, Esrrb, Zfp42, Dppa4, Klf2, Prdm14, Klf4, Tfcp2l1*) from the RNAseq samples of IκBα-WT and IκBα-KO EBs at 96 hr. Normalized counts based on z-score are represented. (**E**) Z- score values for gene sets referring to Endoderm, Mesoderm, and Ectoderm formation from The Gene Ontology database (GO:0001706, GO:0001707, and GO:0001705 terms, respectively) and obtained from corresponding expression levels at mESCs, 48 hr, and 96 hr Ebs IκBα-WT or IκBα-KO cells (three independent clones of each genotype). Loess curves are also represented with 95% confidence intervals (shadowed area surrounding the curves). (**F**) Schematic of teratoma formation assay. 5×10^5 IκBα-WT or IκBα-KO murine Embryonic Stem Cells (mESCs) cultured in Serum/Leukemia Inhibitory Factor (LIF) were injected intramuscularly in the leg of immunocompromised NSG mice. Six weeks after the transplant, teratomas were formed, and mice were euthanized for further teratoma analysis. (**G**) Representative images of immunohistochemistry for OCT3/4 in IκBα- WT (left panel) or IκBα-KO (right panel) teratomas. Teratomas were derived from three independent clones of mESCs of both genotypes. (**H**) Percentage of OCT3/4+ cells in teratomas. Positive cells were counted from five different microscope fields from each IκBα-WT and IκBα- KO clone. Unpaired two-sided t-test was performed. Bars indicate mean values and error bars refer to ± SD. Significance level of 0.05 is considered for all statistical tests.

The online version of this article includes the following source data and figure supplement(s) for figure 2:

**Figure supplement 1.** Depletion of IκBα in murine Embryonic Stem Cells (mESCs) compromises the exit from pluripotency.

**Figure supplement 1—source data 1.** Original files for western blot analysis displayed in *Figure 2—figure supplement 1A*.

**Figure supplement 1—source data 2.** Raw files for western blot analysis displayed in *Figure 2—figure supplement 1A*.

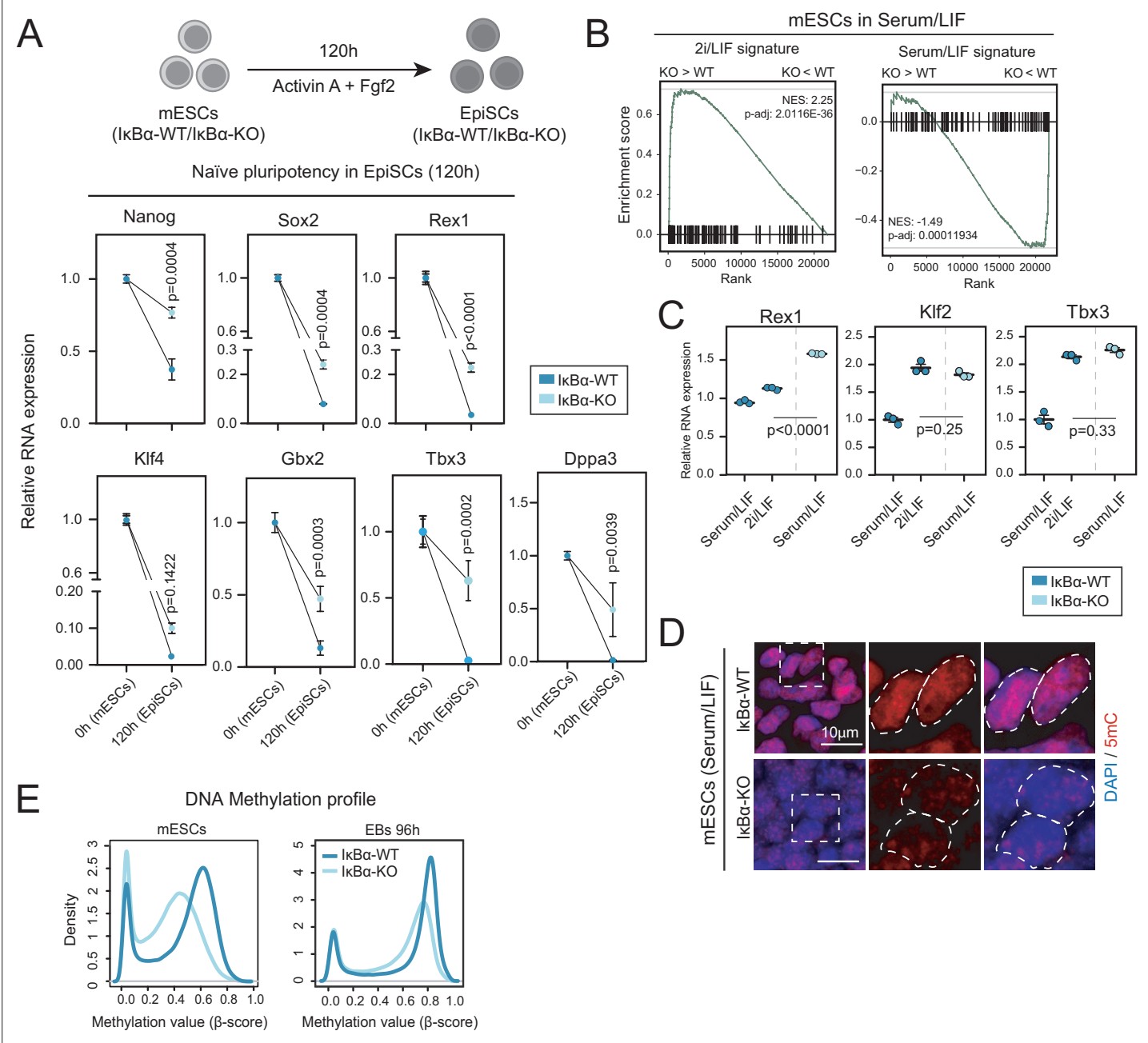

**Figure 3.** IκBα-KO murine Embryonic Stem Cells (mESCs) retain the ground-state of naïve pluripotency under Serum/Leukemia Inhibitory Factor (LIF) culture. (**A**) (Upper panel) Schematics of mESCs differentiation towards Epiblast Stem Cells (EpiSCs). mESCs cultured in Serum/LIF were then seeded in N2B27 medium supplemented with 10 ng/mL Fgf2 and 20 ng/mL Activin A for 120 hr (see Materials and methods section for further information). (Bottom panel) Relative RNA levels (based on qPCR experiments) of naïve pluripotency genes (*Nanog, Sox2, Rex1, Klf4, Gbx2, Tbx3, Dppa3*) in IκBα-WT or IκBα-KO EpiSCs at 120 hr. RNA levels were normalized based on *Tbp* expression relative to IκBα-WT. Data from three independent clones from either IκBα-WT or IκBα-KO genotypes. Unpaired two-sided t-test applied. Dots indicate mean values and error bars refer to ± SD. (**B**) GSEA results comparing IκBα-KO vs IκBα-WT mESCs cultured in Serum/LIF against ground (2i/LIF) and naïve (Serum/LIF) state pluripotency signatures retrieved from *Ghimire et al., 2018*. Adjusted p-value by Benjamini-Hochberg procedure and Normalized Enrichment Score (NES) indicated. (**C**) Relative RNA levels of the naïve pluripotency genes *Rex1 (Zfp42), Klf2*, and *Tbx3* upon culture of IκBα-WT either in Serum/LIF or two passages in 2i/LIF. 2i/LIF IκBα-WT mESCs were compared with IκBα-KO mESCs cultured in Serum/LIF. Unpaired two-sided t-test was applied to calculate statistical significance. Each dot represents an independent clone from each of the two genotypes. Horizontal bars indicate mean values and error bars refer to ± SD. (**D**) Representative immunofluorescence images of DNA-methylation-related mark 5-methylcytosine (5 mC) in IκBα-WT (left) and IκBα-KO (right) mESCs cultured in Serum/LIF. (**E**) DNA methylation profile (using DNA methylation arrays; *Zhou et al., 2022*) in IκBα-WT vs IκBα-KO mESCs (upper) and 96 hr EBs (bottom

*Figure 3 continued on next page*

*Figure 3 continued*

panel) showing the distribution density of mean ß-values from all 261,220 CpGs under test per condition. Significance level of 0.05 is considered for all statistical tests.

The online version of this article includes the following figure supplement(s) for figure 3:

**Figure supplement 1.** Absence of IκBα favors the ground state of naïve pluripotency in Serum/Leukemia Inhibitory Factor (LIF) conditions.

Overall, these data demonstrate that IκBα is required for the exit of the naïve state of pluripotency and its deficiency results in impaired differentiation of mESCs in vivo and in vitro. Accordingly, IκBα-KO mESCs cultured in Serum/LIF showed a genome-wide pattern of DNA hypomethylation.

## Lack of IκBα causes an epigenetic rewiring in pluripotent stem cells to resemble 2i/LIF naïve pluripotency in serum/LIF culture

ChIP Enrichment Analysis (ChEA) of genes identified in our RNA-seq data of IκBα-KO and WT mESCs revealed that differentially expressed genes were putative targets of chromatin regulators, including the PRC2 subunit SUZ12 and MTF2 (*Figure 4A*). We previously described that IκBα is important for the proper deposition of the H3K27me3 mark in different types of tissue stem cells by associating with elements of the PRC2 complex (*Mulero et al., 2013*; *Marruecos et al., 2020*). Since 2i/LIF and Serum/LIF states are characterized by important differences in epigenetic marks, we carried out an epigenetic profiling of IκBα-WT and IκBα-KO mESCs cultured in Serum/LIF. We performed chromatin immunoprecipitation followed by sequencing (ChIP-seq) of histone modifications associated with gene activation (H3K4me3) and gene repression (H3K27me3) (*Figure 4B* and *Figure 4—figure supplement 1A and B*). Overall, Serum/LIF IκBα-KO mESCs exhibited a general increased H3K4me3 status and redistribution of the PRC2-catalyzed H3K27me3 mark (*Figure 4B*). Notably, gain of the H3K4me3 mark was associated with a reduction of H3K27me3 levels in genes associated with naïve pluripotency function (*Figure 4B* and *Figure 4—figure supplement 1A and B*). In particular, 41 genes with concomitant gain of H3K4me3 and loss of H3K27me3 were found to be enriched in mechanisms associated with pluripotency function (*Figure 4C*), including Tbx3 and Gbx2 (*Figure 4D–E*). A second set of pluripotency-related genes showed an increase in H3K4me3 without changes in H3K27me3 (Zfp42 or Klf2). Both sets of pluripotency genes showed increased expression in the absence of IκBα (*Figure 4H*).

We next aimed to study whether the lack of IκBα was also affecting distal regulatory regions by analyzing enhancer-associated histone marks in IκBα-WT and IκBα-KO mESCs. We quantified the total amount of H3K27Ac (active enhancer) and H3K4me1 (poised enhancer) by ChIP-seq experiments conducted in both IκBα-WT and IκBα-KO mESCs. Overall, IκBα-KO mESCs showed a reduction in poised enhancers at genes (or related genomic regions) associated with differentiation processes (*Figure 4F* and *Figure 4—figure supplement 1C*) and an increase in active enhancers at genes associated with pluripotency (*Figure 4F* and *Figure 4—figure supplement 1D*), such as Tbx3, Tfcp2l1, and Zfp42 (Rex1) (*Figure 4G*), which positively favors their higher expression in IκBα-KO mESCs (*Figure 4H*). These results further support that mESCs lacking IκBα are epigenetically remodeled to favor the ground state of naive pluripotency, which may negatively impact on their differentiation potential, similar to that described for other models with stabilized naïve pluripotent state (*Choi et al., 2017*; *Di Stefano et al., 2019*).

## The chromatin function of IκBα in pluripotency exit is independent of classical NF-κB activity

Although previous results from *C. elegans* (*Brena et al., 2020*) and *Drosophila* (*Mulero et al., 2013*) support the functional relevance of chromatin-related IκBα function, the investigation of this alternative IκBα activity remains challenging due to the predominant role of this protein in NF-κB regulation. However, and similar to that found in other models, we did not detect major changes in Canonical NF-κB target genes upon IκBα depletion neither in Serum/LIF mESCs, nor 48 hr and 96 hr EB-differentiating conditions (*Figure 5A*). Thus, either NF-κB regulation does not require IκBα at this stage or it might be compensated by the other IκBs (IκBβ, IκBε, or p100), as previously demonstrated (*Hayden and Ghosh, 2008*; *Basak et al., 2007*).

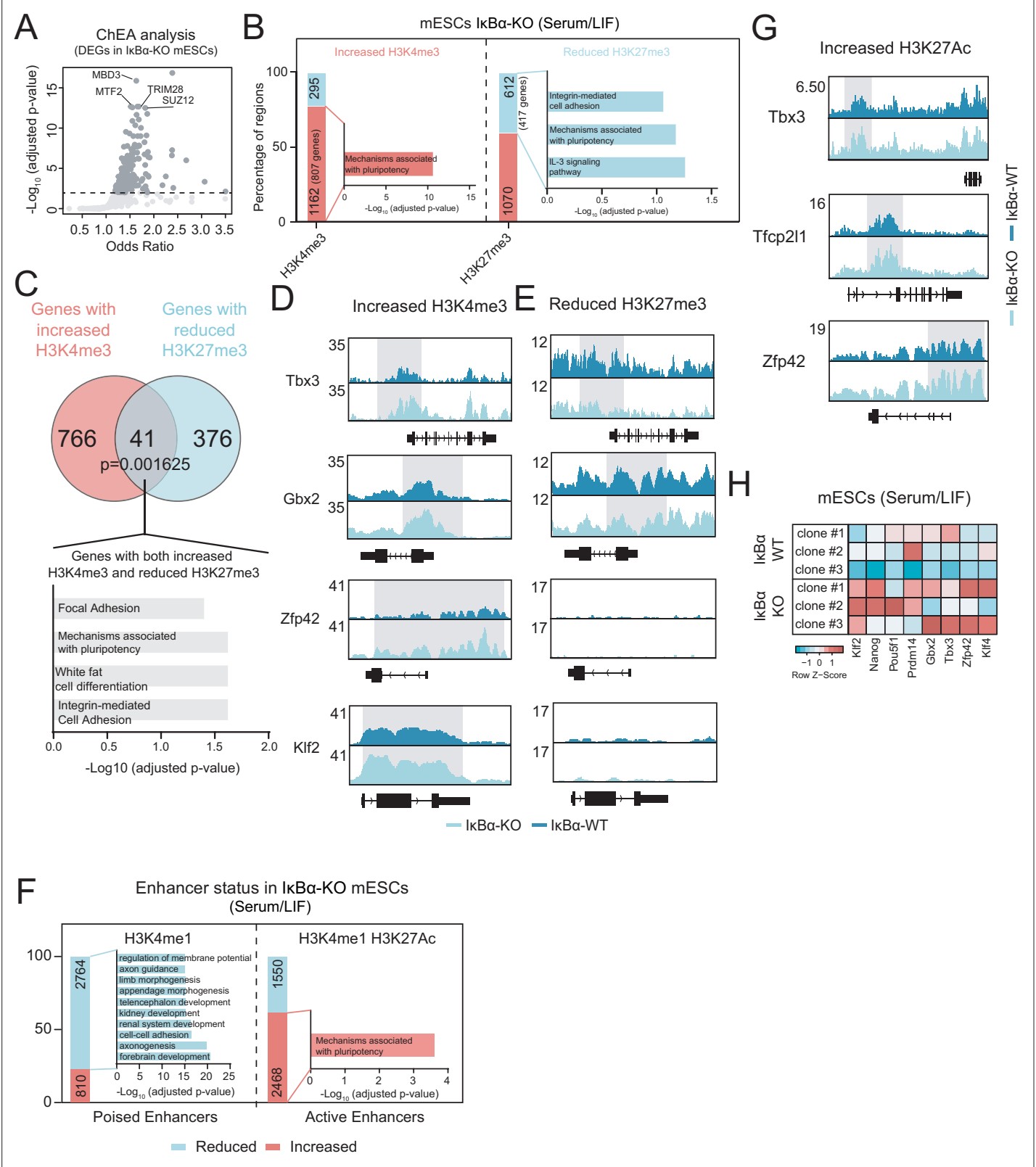

**Figure 4.** Absence of IκBα reinforces the pluripotency program at the epigenetic and transcriptional levels. (**A**) ChIP-seq Enrichment Analysis (ChEA) of differentially expressed genes (DEGs; adjusted p-value <0.05) in IκBα-KO murine Embryonic Stem Cells (mESCs). P-value was calculated using Fisher exact test. (**B**) Number of differentially bound regions (adjusted p-value <0.05, FDR) for H3K4me3 and H3K27me3 histone marks in IκBα $^{-/-}$ vs IκBα $^{+/+}$ mESCs cultured in Serum/Leukemia Inhibitory Factor (LIF). Enriched or reduced regions in IκBα $^{-/-}$ mESCs are distinguished (log$_2$ Fold-change >0

*Figure 4 continued on next page*

*Figure 4 continued*

or <0, respectively). Overrepresented WikiPathways, for (left) H3K4me3 and (right) H3K27me3 are indicated. Annotated genes to differential regions showing H3K4me3 enrichment or H3K27me3 reduction were considered. A one-sided hypergeometric test was conducted. (**C**) Venn Diagram of genes contained in regions with either increased H3K4me3 or reduced levels of H3K27me3. Statistical significance of the overlapping was calculated using the chi-squared method. Overrepresented pathways for genes containing increased H3K4me3 and reduced H3K27me3 levels are indicated (only pathways with an adjusted p-value <0.05 were considered). (**D–E**) Representative genomic regions of naïve pluripotency genes having either differential enriched H3K4me3-only (**D**) or reduced H3K27me3-only (**E**) levels in IκBα-KO vs IκBα-WT mESCs cultured in Serum/LIF. Shadowed regions highlight the differential levels of the histone marks. (**F**) Number of regions with differential enhancer activity in IκBα-KO vs IκBα- WT mESCs cultured in Serum/LIF. Increased/Reduced activity for (left) poised enhancers based on the H3K4me1 differential binding and absence of H3K27ac overlapping peaks in IκBα-KO mESCs and (right) active enhancers based on the differential binding of H3K27ac and presence of H3K4me1 overlapping peaks in IκBα-KO. Differential binding based on adjusted p-value <0.05 (FDR). Increased or reduced regions in IκBα-KO are distinguished by log$_2$ Fold-change>0 or<0, respectively, compared to IκBα-WT mESCs. Overrepresented WikiPathways for annotated genes to (left) poised enhancers with reduced activity and (right) active enhancers with increased activity in IκBα-KO vs IκBα-WT mESCs are indicated (adjusted p-value <0.05, FDR). A one-sided hypergeometric test was conducted. (**G**) Representative genomic regions of naïve pluripotency genes having an increase in active enhancer status (H3K27Ac enrichment) in IκBα-KO mESCs cultured in Serum/LIF. Shadowed regions highlight the differential levels of the histone mark. (**H**) Heatmap showing the expression levels of the naïve pluripotency genes from the RNAseq samples of IκBα-KO and IκBα-WT mESCs cultured in Serum/LIF. Normalized counts (z-score) from genes are represented.

The online version of this article includes the following figure supplement(s) for figure 4:

**Figure supplement 1.** Epigenetic rewiring characterization in IκBα-KO murine Embryonic Stem Cells (mESCs) cultured in Serum/Leukemia Inhibitory Factor (LIF).

To further investigate the relative impact of both IκBα functions in mESCs, we took advantage of recent results from our group that led to the identification of the specific IκBα protein residues that define NF-κB- or chromatin-binding through H2A/H4. Mutation of these residues allowed the generation of doxycycline-inducible (i) separation-of-function IκBα mutants that are specifically deficient in one or the other function, at least in the intestinal epithelial cells. We called i-IκBα$^{\Delta Chromatin}$ the i-SOF IκBα mutant that interacts with NF-κB but is deficient for H2A and H4 binding and i-IκBα$^{\Delta NF-\kappa B}$ the one that interact with histones but is deficient for NF-κB subunits binding (**Figure 5B** and **Álvarez-Villanueva et al., 2023**). The three different IκBα forms (i- IκBα$^{WT}$, i-IκBα$^{\Delta NF-\kappa B}$, and i-IκBα$^{\Delta Chromatin}$) were stably transfected in the IκBα KO mESCs and induced by 16 hr of doxycycline treatment (**Figure 5—figure supplement 1A**). We used these cells to investigate the biochemical properties and functional impact of SOF IκBα mutants in mESCs. By pull-down experiments, we found that i-IκBα$^{\Delta NF-\kappa B}$ expressed in ESCs was able to interact with histone H2 A but not with the NF-κB member p50. In contrast, i-IκBα$^{\Delta Chromatin}$ interacts with p50 but not with H2A, while i-IκBα$^{WT}$ preserves both binding capacities (**Figure 5C**). In concordance with these results, i-IκBα$^{WT}$ and i-IκBα$^{\Delta NF-\kappa B}$ were found in both the cytoplasmic and chromatin fractions of mESCs, whereas IκBα$^{\Delta Cromatin}$ was retained in the cytoplasm (**Figure 5D–E**). In multiple experiments, we noticed a significant reduction in protein levels of IκBα$^{\Delta chromatin}$ despite the mRNA expression levels of the different IκBα forms being comparable (**Figure 5C and D** and **Figure 5—figure supplement 1A**).

Then, we studied whether i-IκBα$^{WT}$ and/or the i-SOF IκBα mutants (IκBα$^{\Delta NF-\kappa B}$ or IκBα$^{\Delta Chromatin}$) were able to reverse any of the phenotypes observed in the IκBα-KO mESCs (**Figure 5—figure supplement 1B**). Induction of i-IκBα$^{WT}$ and i-IκBα$^{\Delta NF-\kappa B}$ reduced the expression levels of the naïve pluripotent genes *Zfp42, Klf2, Sox2,* and *Tbx3*. On the other hand, the same genes either do not change their expression (*Zfp42, Sox2, Klf2*) or increase their levels (*Tbx3*) upon i-IκBα$^{\Delta Chromatin}$ induction (**Figure 5F**). In addition, the ground state-associated DNA hypomethylation was specifically reverted in i-IκBα$^{WT}$ and i-IκBα$^{\Delta NF-\kappa B}$ mESCs but not in i-IκBα$^{\Delta Chromatin}$ mESCs (**Figure 5G**). Finally, we addressed whether i-IκBα$^{WT}$, i-IκBα$^{\Delta NF-\kappa B}$, and i-IκBα$^{\Delta Chromatin}$ mESCs were capable to revert the differentiation blockage of IκBα-KO ESCs (**Figure 5—figure supplement 1B**). Induction of IκBα$^{WT}$ and IκBα$^{\Delta NF-\kappa B}$ was enough to restore the differentiation potential of IκBα-KO mESCs into EBs, as indicated the increase in the number of differentiated cells (based on percentage of SSEA-1$^{neg}$ cells) (**Figure 5H** and **Figure 5—figure supplement 1C**) and the reduced number of AP + colonies (in the 216 hr IκBα-KO reconstituted EB assays), which was similar to the number of colonies obtained from IκBαWT EBs (**Figure 5I**). Lastly, the number of remaining pluripotent-like cells (defined by OCT3/4 NANOG staining) was also reduced in i-IκBα$^{WT}$ and i-IκBα$^{\Delta NF-\kappa B}$ 216 h EBs, and only small and few regions of undifferentiated cells was observed. In contrast, induction of i- IκBα$^{\Delta Chromatin}$ did not restore the differentiation potential of IκBα-KO mESCs (**Figure 5J** and **Figure 5—figure supplement 1D**).

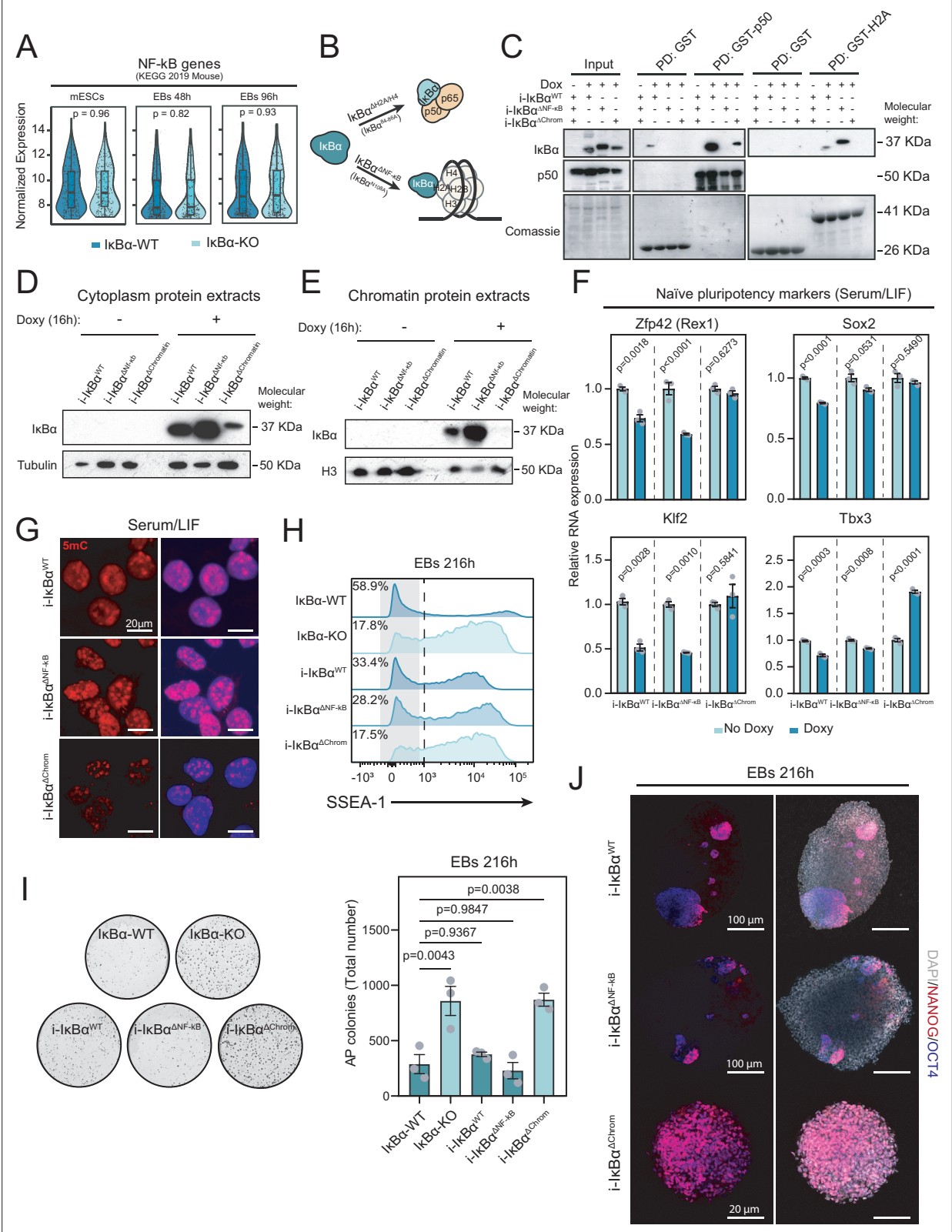

**Figure 5.** IκBα prevents the hyperactivation of naïve pluripotency program in an NF-κB- independent manner. (**A**) Boxplots, violin plots, and individual values representing the normalized gene expression levels in three independent clones of IκBα-WT or IκBα-KO mESCs, embryoid bodies (EBs) 48 hr, and EBs 96 hr obtained from bulk RNAseq data. Considered genes are those annotated to NF-κB signaling pathway (KEGG PATHWAY mmu04064). Box Plots: center line, median; lower and upper hinges, first and third quartiles; whiskers, 1.5x interquartile range (IQR) extended to the largest/smallest

*Figure 5 continued*

value within 1.5x IQR; no outliers present. Unpaired two-sided Wilcoxon test applied. (**B**) Schematic of chromatin- specific/NF-κB-deficient (IκBα$^{\Delta NF-\kappa B}$) or NF-κB-specific/chromatin-deficient (IκBα$^{\Delta Chrom}$) IκBα mutants (Separation-Of-Function or SOF mutants). (**C**) Pull-down of GST-H2A and GST-P50 with cell lysates from different SOF IκBα mutant mESCs. (**D**) Western blot analysis of cytoplasmic protein extracts of doxycycline-inducible IκBα forms (i-IκBα$^{\Delta WT}$, i-IκBα$^{\Delta NF-\kappa B}$, i-IκBα$^{\Delta Chrom}$) after 16 hr of doxycycline induction. (**E**) Western blot analysis of chromatin protein extracts of oxycycline-inducible IκBα forms (i-IκBα$^{\Delta WT}$, i-IκBα$^{\Delta NF-\kappa B}$, IκBα$^{\Delta Chrom}$) after 16 hr of doxycycline induction. (**F**) Relative RNA levels of naïve pluripotency genes (*Zfp42, Klf2, Sox2, Tbx3*) in i- IκBα$^{WT}$, i-IκBα$^{\Delta NF-\kappa B}$, and i-IκBα$^{\Delta Chrom}$ mESCs after doxycycline induction. Unpaired two-sided t-test was performed. Dots indicate three independent experiments. Bars indicate mean values and error bars refer to ± SEM. (**G**) Representative immunofluorescence images of DNA-methylation-related mark 5-methylcytosine (5 mC) in i-IκBα$^{\Delta WT}$ (upper panel), i-IκBα$^{\Delta NF-\kappa B}$ (medium panel), IκBα$^{\Delta Chrom}$ (bottom panel) murine Embryonic Stem Cells (mESCs) cultured in Serum/Leukemia Inhibitory Factor (LI)F. (**H**) Flow cytometry analysis of the pluripotency surface marker SSEA-1 at 216 hr embryoid bodies (EBs) in i-IκBα$^{WT}$, i-IκBα$^{\Delta NF-\kappa B}$ and i-IκBα$^{\Delta Chrom}$ cells. (**I**) Quantification of Alkaline Phosphatase (AP) staining in 216 hr EBs upon reconstitution with the different IκBα forms. Each dot represents an independent experiment. One-way ANOVA was applied followed by Tukey's post-hoc test for all pairwise comparisons. Bars indicate mean values and error bars refer to ± SEM. Representative images of Alkaline Phosphatase (AP) staining in 216 hr EBs are included in the upper panel. (**J**) Representative immunofluorescence for NANOG and OCT3/4 at 216 hr EBs in i-IκBα$^{WT}$ (upper panel), i-IκBα$^{\Delta NF-\kappa B}$ (medium panel), and i-IκBα$^{\Delta Chrom}$ (bottom panel) cells. Significance level of 0.05 is considered for all statistical tests.

The online version of this article includes the following source data and figure supplement(s) for figure 5:

**Source data 1.** Annotated files for western blot analysis displayed in *Figure 5C, D, E*.

**Source data 2.** Raw files for western blot displayed in *Figure 5C, D, E*.

**Figure supplement 1.** Reversion of IκBα-KO-related hyper pluripotent status is NF-κB independent.

These results demonstrate that the chromatin-dependent IκBα function, but not its NF-κB- related activity, is necessary for the proper regulation of chromatin marks at specific genomic regions that are linked to transcriptional changes that are likely at the base of the capacity of ESCs to exit the ground state of naïve pluripotency and differentiate.

## Discussion

Here, we have identified a new and unexpected function for the IκBα protein in the regulation of the naïve pluripotency exit. Using mESCs to investigate the role of IκBα in pluripotency maintenance and differentiation, we discovered that the absence of IκBα severely impairs the exit of pluripotency. Furthermore, IκBα-KO mESCs stabilizes the ground state of naïve pluripotency under Serum/LIF conditions, shedding light on its function as a critical regulator of the transition from naïve to primed pluripotency. While IκBα is commonly known for its role as an inhibitor of the NF-κB signaling pathway (*Zhang et al., 2017*), our laboratory has previously furnished compelling evidence of its alternative role through its interaction with histones and other chromatin components (*Mulero et al., 2013*; *Marruecos et al., 2021*; *Aguilera et al., 2004*). We have demonstrated that chromatin-bound IκBα plays a crucial role in regulating skin and intestinal stem cells (*Mulero et al., 2013*; *Marruecos et al., 2020*), hematopoietic stem cell development (*Thambyrajah et al., 2022*), and it influences the regenerative capacity of these tissues, and in some cases, their susceptibility to neoplastic transformation. Disentangling the chromatin-related function of IκBα from its canonical role as an inhibitor of NF-κB has been challenging, making it difficult to determine its truly biological significance. To tackle this challenge, we engineered separation-of-function (SOF) mutants of IκBα, as detailed in *Álvarez-Villanueva et al., 2023*, by identifying specific residues that are essential for binding either NF-κB elements or histones. Interestingly, we have shown that only the histone-binding proficient IκBα mutant (IκBα$^{\Delta NF-\kappa B}$), which lacks the ability to bind to NF-κB, has the capability to reverse the stabilization of the ground state observed in IκBα-KO mESCs. This is sufficient to facilitate their exit from the state of naïve pluripotency, restoring their full differentiation potential (*Figure 5*). The involvement of NF-κB in pluripotency has been a subject of prior investigation. Studies have revealed that pluripotent cells display a dampened inflammatory signaling pathway, which is induced upon pluripotency dissolution and activation of differentiation (*Torres and Watt, 2008*; *Guo, 2019*). However, the precise molecular mechanisms underlying the attenuated status of NF-κB during this stage remain incompletely understood. A recent work has shed light on a potential regulatory axis involving ATG5-β- TrCP1-NF-κB, proposing it as a mechanism that diminishes NF-κB activity by stabilizing the IκBα protein in mESCs (*Li et al., 2024*). In any case, since modification of canonical NF-κB activity directly impacts on the levels IκBα expression, the previous NF-κB-associated phenotypes should be reevaluated considering this

chromatin role of IκBα. Furthermore, the expression of IκBα$^{\Delta Chromatin}$, which retains the inhibitory function of NF-κB, failed to mitigate the ground state stabilization observed in IκBα-KO mESCs (*Figure 5*). This finding suggests that the observed role of IκBα in regulating the exit from naïve pluripotency is not due to NF-κB inhibition. In fact, there is evidence that the chromatin function of IκBα is an ancestral function preceding its role as NF-κB inhibitor. In *Caenorhabditis elegans* (*C. elegans*), which lacks NF-κB orthologs but possesses two homologs of IκBα, the absence of IκBα orthologs has severe multi-organ differentiation defects (*Brena et al., 2020*). This observation further supports the notion that the chromatin-related function of IκBα plays a fundamental role in developmental processes, independent of its canonical role in NF-κB signaling. Nonetheless, differences in the expression level of SOF IκBα mutants, with the IκBα$^{\Delta Chromatin}$ expressed at lower protein levels compared to the IκBα$^{\Delta NF-\kappa B}$, may also impact in the rescue phenotype.

We have observed that IκBα-KO mESCs exhibit epigenetic and transcriptomic profiles resembling the ground state of naïve pluripotency under Serum/LIF culture (*Figure 3*). In this regard, the fact that IκBα is preferentially located in the chromatin of mESCs cultured in Serum/LIF, when they exhibit a high degree of heterogeneity, suggests that IκBα might act as a regulator of this heterogeneity. One potential explanation might be that IκBα could be affecting the stability of fluctuating transcription factors, which is crucial to mediate the pluripotent-to-differentiation balance (*Martinez Arias and Brickman, 2011*). Thus, the lack of IκBα might stabilize those transcription factors that would favor the ground-state observed in IκBα-KO mESCs cultured in Serum/LIF. Although IκBα KO mESCs exhibit a transcriptional phenotype and hypomethylation state that resembles the ground state of naïve pluripotency, there are only modest changes on histone marks associated to enhancers (H3K27Ac) or gene regulation (H3K4me3 and H3K27me3). Altogether indicates that further experiments are required to fully elucidate the effect of chromatin IκBα. Moreover, the differences in murine and human pluripotency (*Weinberger et al., 2016*) point out the necessity to address the real impact of IκBα modulation in the stabilization of the ground state in human pluripotent stem cells.

In summary, our work establishes that IκBα mediates the exit from naïve pluripotency by modulating the activation of naïve pluripotency genes. Importantly, this newly identified role of IκBα operates independently of its canonical function in inhibiting the NF-κB pathway. Our findings underscore the intricate interplay between inflammation and pluripotency status, revealing a previously unrecognized complexity in their interaction.

## Methods

### Cell culture

mESC line ES-E14TG2a (ATCC; Cat #CRL-1821) was cultured on plastic dishes precoated with 0.1% (w/v) Gelatin (Sigma; Cat #G2500-100G) in Serum/LIF medium. Serum/LIF is composed of DMEM basal medium (Sigma-Aldrich; Cat #D5796) supplemented with 15% FBS (ESC- qualified; Gibco Cat #26140079), 1X Glutamax (Gibco; Cat #35050061), 1X NEAA (Gibco; Cat#11140050), 1 mM Sodium Pyruvate (Gibco; Cat#11360070), 1000 U/mL Leukemia Inhibitory Factor (LIF) (Millipore, Cat #ESG1107), and 0.125 mM 2-mercaptoethanol (Gibco; Cat#31350010). Medium was changed every day, and cells were splitted every other day using TrypLE Express (Gibco; Cat #12605010) for harvesting. Cells were maintained in a 5% $CO_2$ incubator at 37°C.

For inducing the ground-state of naïve pluripotency, mESCs were cultured in 2i/LIF medium for two consecutive passages. 2i/LIF medium is composed of NDiff 227 medium (TAKARA, Cat #Y40002) supplemented with 1000 U/mL LIF, 0.4 µM PD032591 (Selleck Chemicals, Cat #S1036), and 3 µM CHIR99021 (Merck, Cat #SML1046).

### Epiblast stem cell differentiation

Differentiation from mESCs towards Epiblast Stem Cells (EpiSCs) was performed as previously described (*Pantier et al., 2019*). 3×10$^4$ mESCs were seeded in 6-well plastic dishes, and they were cultured in Serum/LIF medium for 24 hr. Medium was then switched to NDiff 227 medium (TAKARA, Cat #Y40002) supplemented with 20 ng/ml activin A (Cat. #120-14E; PeproTech) and 10 ng/ml Fgf basic (R&D Systems, Cat #233-FB-025/CF). Cells were submitted to daily media changes till day 5 (120 hr), when they were further analyzed.

## Embryoid bodies differentiation from mESCs

Embryoid bodies differentiation was established as described in *Sroczynska et al., 2009*. Briefly, mESCs were splitted twice in Serum/LIF medium before inducing differentiation. Once mESCs are 80% confluent, cells were collected using TrypLE Express. One well of 6-well plate was splitted in an entire 6-well plate in IMDM-ES medium for 48 hr. IMDM-ES medium is composed of Iscove's Modified Dulbecco's Medium (IMDM) (Cytiva, Cat #16SH30259.01) supplemented with 20% FBS (ESC-qualified; Gibco Cat #26140079), 1X Glutamax (Gibco; Cat #35050061), 1X NEAA (Gibco; Cat#11140050), 1 mM Sodium Pyruvate (Gibco; Cat#11360070), 10 ng/mL Leukemia Inhibitory Factor (LIF) (Millipore, Cat #ESG1107), and 0.125 mM 2-mercaptoethanol (Gibco; Cat#31350010). For embryoid body induction, mESCs were harvested, and they were rinsed twice with DPBS (Gibco; Cat #14190144). Cells were very well disaggregated into single-cells, and $1.2 \times 10^4$ cells/mL embryoid body differentiation (EB$^{diff}$) medium were resuspended to a total volume of 25 mL of EB$^{diff}$ medium. EB$^{diff}$ medium is composed of IMDM supplemented with 15% FBS (ESC-qualified; Gibco Cat #26140079), 1X Glutamax (Gibco; Cat #35050061), 50 µg/mL ascorbic acid (Sigma; Cat #A-4544), 180 µg/mL Transferrin (Roche; Cat #10652202001), and 0.45 mM alpha-monothioglycerol (MTG) (Sigma; Cat #M6145). EBs were formed in suspension for 5 days (120 hr). At day 5, in order to elongate the differentiation up to 216 hr (day 9), EB$^{diff}$ medium was refreshed by harvesting EBs with 10 mL serological pipette, centrifuge at 200 g for 3 min and EBs were resuspended in fresh EB$^{diff}$ medium for four more days.

## CRISPR/Cas9 gene editing and cell lines generation in mESCs

The two guide RNAs (gRNAs) targeting the Nfkbia locus were designed using the CRISPR design tool from MIT (http://crispr.mit.edu). The best 2 gRNAs (based on on-target and off-target scores) that were targeting the exon 1 of Nfkbia (IκBα gene) were selected. After annealing, one of the gRNAs was cloned into SpCas9(BB)–2A-GFP (px458) plasmid (Addgene; Cat #48138), and the other gRNA was cloned into the px330-mCherry plasmid (modified from Addgene; Cat #98750 to incorporate mCherry reporter). mESCs were co-transfected with the two plasmids (px458-gRNA1 and px330-mCherry-gRNA2). $3 \times 10^5$ cells were seeded a day before per well of 6-well plate. At day of transfection, cells were washed once with 1 X DPBS (Gibco; Cat #14190144), and 2 mL Opti-MEM medium (Gibco; Cat #31985070) was added into cells. 1.25 µg of each plasmid (2.5 µg of total DNA) was incubated with 10 µL of Lipofectamine 2000 (Invitroge Cat #11668019) in 240 µL of Opti-MEM for 20 min. OptiMEM::Lipofectamine::DNA mixture was added into mESCs, and cells were incubated for 5 hr. Cells were then washed with DPBS, and medium was replaced by Serum/LIF. 48 hr after transfection, GFP$^+$ mCherry$^+$ single cells were sorted by FACS using BD FACSAria II Cell Sorter (BD Bioscience). Mutant clones were screened through PCR and western blotting to identify single clones with no IκBα protein expression. Non-targeting scrambled gRNAs were cloned into px458 and px330-mCherry plasmids as wild-type clones. Three independent clones from each genotype (IκBα-WT and IκBα-KO) were selected for further experiments.

Doxycycline-inducible IκBα (IκBα$^{WT}$, IκBα$^{\Delta NF-\kappa B}$, and i-IκBα$^{\Delta H2A/H4}$) mESCs were generated by cloning the three different IκBα versions into a PiggyBac transposon system (*Chen et al., 2020*). IκBα protein-coding cDNAs were PCR-amplified, and NheI and SalI restriction sites were placed at the 5' and 3' ends, respectively. T2A-EGFP fragment was amplified from PX458 plasmid, and SalI and AgeI restriction sites were introduced at its 5' and 3' ends. PB-TRE backbone (Addgene, Cat# 63800) was digested with NheI and AgeI enzymes, and mNfkbia and T2A-EGFP fragments were ligated into the PB-TRE digested vector. To generate mESCs containing the inducible vector, IκBα-KO mESCs were co-transfected with PB-TRE-mNfkbia-T2A-EGFP and PiggyBac transposase plasmids, and pool of transfected cells was selected by Hygromycin (100 µg/mL) for 7 days. Cells were doubly screened by FACS sorting green fluorescent protein after 24 hr of doxycycline (1 µg/mL) treatment. Pool of cells were used for further experiments. Oligos sequences are included in *Supplementary file 1*.

## Teratoma formation assay

IκBα-WT and IκBα-KO mESCs grown in Serum/LIF ($5 \times 10^5$) were injected intramuscularly into severe combined immunodeficient mice (NSG). In order to favor 3D aggregation of cells and teratoma formation, cells were resuspended in Matrigel Matrix (BD; Cat #356234) prior to injection. Six weeks later, mice with tumors were euthanized, and tumors were fixed in formaldehyde, embedded in paraffin, sectioned, and stained with hematoxylin and eosin for histological analysis.

## Immunofluorescence staining

For immunostaining of mESCs, cells were seeded on 0.1% gelatin-coated coverslips on 6-well plates; for EB immunofluorescence, they were collected directly from the plate. Samples were washed twice with 1X PBS, and they were fixed at 4°C for 30 min with 4% PFA (Electron Microscopy Sciences; Cat #15713S). For immunofluorescence of 5-Methylcytosine (5 mC), fixed cells were incubated with 2M Hydrochloric Acid (Sigma-Aldrich, Cat #H1758) for 20 min at room temperature. Samples were then washed twice with 1X Tris-buffered saline (TBS; 50 mM Tris-Cl, pH 8) and permeabilized and blocked with 1X TBS supplemented with 1% Triton-X100 (MERCK; Cat #9036-19-5) and 6% FBS (Biological Industries; Cat #04-001-1A) for 2 hr at 4°C. Samples were washed twice with 1X TBS supplemented with 6% FBS. Primary antibody incubation was performed in 1X TBS plus 6% FBS and 0.3% or 0.5% Triton-X100 (0.3% for 2D culture and 0.5% for EBs) overnight at 4°C. The following primary antibodies were used: OCT3/4 (1:250; Santa Cruz; Cat #sc-5279), NANOG (1:250; Novus Biologicals; Cat #NB100- 588), 5-Methylcytosine (1:500, Invitrogen, Cat #MA5-24694), H3K27Ac (1:2000, Abcam, Cat #AB4729). Samples were then washed four times with 1X TBS, 5 min each washing. Secondary antibody incubation was performed in TBS 1X plus 1% BSA (Sigma-Aldrich; Cat #9048-46-8) or 2 hr at room temperature. The following antibodies were used: Alexa Fluor 488 donkey anti-mouse antibody (1:1000; Invitrogen; Cat #A-21202), Alexa Fluor 647 donkey anti-mouse antibody (1:1000; Invitrogen; Cat #A-31571), Alexa Fluor 594 donkey anti-rabbit antibody (1:1000; Life Technologies; Cat #A-21207), Alexa Fluor 647 donkey anti-rabbit antibody (1:1000, Invitrogen; Cat #A-31573). Samples were washed three times with 1X TBS for 5 min at room temperature after each rinse. Samples were mounted using DAPI Fluoromount-G (Southern Biotech; Cat #0100–20).

## Microscopy and image acquisition

Fluorescence images were acquired using Confocal Leica TCS SP5 (Leica Microsystems), and Leica application software LAS AF (Leica Microsystems) was used to visualize the images. Images of teratoma haematoxylin and eosin and immunohistochemistry stainings were acquired using BX61 Olympus Microscope (Olympus), and PRECiV 2D Image and measurement Software (Olympus) was used to visualize the images. The Image J (version 1.15) was used for further analysis (*Schneider et al., 2012*).

## Flow cytometry sample preparation and analysis

Cells were collected and they were purified based on FACS using BD FACSAria II Cell Sorter (BD Bioscience). Cells were sorted at 3500 events/s, and at maximum flow rate of 4 and 85 µm nozzle. 5 µg/ml of DAPI (Biotium; Cat #BT-40043) was used as a viability dye. For flow cytometry analysis of pluripotency exit, cells were disaggregated using TrypLE Express, and they were incubated at 4°C for 20 min with SSEA1-eFluor 660 Monoclonal Antibody (1:200, Invitrogen, Cat #50-8813-42). BD LSRFortessa Cell Analyzer or BD LSR II Flow Cytometer (BD Bioscience) were used for Flow Cytometry Analysis experiments. Flow cytometry data were analyzed using FlowJo X v10.0.7 (BD Biosciences).

## RNA isolation, cDNA synthesis, and quantitative RT-PCR

Total RNA isolation from cells was performed using the RNeasy Plus Mini Kit (Qiagen; Cat #74136) or RNeasy Micro Kit (Qiagen; Cat #74004) following the manufacturer's instructions. Amount of RNA was quantified with Nanodrop (Thermo Fisher; Cat #ND2000CLAPTOP), and 2 µg of total RNA was retro-transcribed using Transcriptor First Strand cDNA Synthesis Kit (Roche; Cat #04897030001) following the manufacturer's instructions.

Quantitative RT-PCR was performed in triplicates for each sample, and SYBR Green I Master Kit (Roche; Cat #04887352001) was used to carry out the reaction. qRT-PCR was performed using the LightCycler 480 system (Roche). Relative expression levels were calculated as $2^{-\Delta CT}$ normalized with the average CT of the housekeeping gene Tbp or Gapdh. Oligos sequences are found in *Supplementary file 1*.

## Chromatin immunoprecipitation (ChIP)

$4–6\times10^7$ mESCs were cross-linked by incubating them in DPBS (Gibco, Cat #14190094) supplemented with 1% formaldehyde (Sigma-Aldrich; Cat #252549) for 10 min rocking at room temperature. Cross-linking was stopped using 125 mM Glycine (Sigma-Aldrich, Cat #G8790) rocking for 5 min at room temperature. Fixed cells were then washed twice with ice-cold DPBS, and scrapped and

collected using DPBS supplemented with protease/phosphatase inhibitor cocktail composed of 1X cOmplete, EDTA-free Protease Inhibitor Cocktail (Roche, Cat #11873580001), 1 mM PMSF, 1 mM Sodium Orthovanadate and 20 mM β-Glycerol phosphate. Cells were centrifuged at 3200 g for 5 min at 4°C. Cells were lysed by resuspending cell pellet in ice-cold ChIP buffer (1 volume of SDS buffer [100 mM NaCl, 50 mM Tris-HCl pH 8.1, 5 mM EDTA pH 8, 0.5% SDS] and 0.5 volume of Triton dilution buffer [100 mM Tris-HCl pH 8.6, 100 mM NaCl, 5 mM EDTA pH 8, 5% Triton X-100]) supplemented with protease/phosphatase inhibitor cocktail, and samples were sonicated using sonication beads (Diagenode; Cat #C01020031) and the Bioruptor Pico Sonicator (Diagenode; Cat #B01060010) for 20 cycles (each cycle 30 s on/30 s off) or till DNA fragments have 100–300 bp size. Samples were centrifuged at 16,000 g for 20 minutes at 4°C. Chromatin was quantified, and 30 µg of chromatin was incubated with every 5 µg of antibody rotating for 16 hr (or overnight) at 4°C. The following anti-bodies were used: H3K4me3 (Abcam, Cat #ab8580), H3K27me3 (Millipore, Cat ##07–449), H3K27Ac (Abcam, Cat #ab4729), H3K4me1 (Abcam, Cat #ab8895). Samples were pulled down by incubation with Protein A-Sepharose CL-4B (previously hydrated and blocked with 0.05% BSA) for 3 hr at 4°C in rotation. Sample was washed with a low salt washing buffer (50 mM HEPES pH 7.5, 140 mM NaCl, and 1% Triton) three times, one wash with a high salt wash buffer (50 mM HEPES pH 7.5, 500 mM NaCl, and 1% Triton) and one wash with TE pH 8 (10mM Tris-HCl pH8, 1 mM EDTA). Samples were then eluted in 1% SDS and 100 mM NaHCO$_3$. Samples were descrosslinked by incubating for 16 hr (or overnight) at 65°C shaking at 450 rpm, and were treated with Proteinase K for 1 hr at 45°C. DNA was purified using QIAquick PCR Purification Kit (Qiagen, Cat #28106), following the manufacturer's instructions.

For ChIP followed by sequencing (ChIP-seq), purified DNA concentration and integrity were determined using Agilent Bioanalyzer (Agilent Technologies; Cat #G2939BA). Libraries were prepared using standard protocols. Chromatin was sequenced using Illumina HiSeq platform (Illumina, Inc) (50 bp single-end reads). Samples sequencing depth range was the following: (i) H3K4me3: 2.56–2.74×10$^7$ reads, (ii) H3K27me3: 6.19–7.2×10$^7$ reads, (iii) H3K27Ac: 9.11–10.65×10$^7$ reads and (iv) H3K4me1: 9.76–10.58×10$^7$ reads.

## ChIP-seq data analysis

Quality control was performed on raw data with FASTQC tool. Raw reads were trimmed to remove adapters presence with Trimgalore (v0.6.6) (*Ewels and Afyounian, 2012*). Default parameters were used except for a minimum quality of 15 (Phred score) and an adapter removal stringency of 3 bp overlap. For H3K4me3, H3K27me3, and H3K4me1 data, trimmed reads were aligned to the reference genome with Bowtie2 (v2.4.4) which was executed with default parameters (*Langmead and Salzberg, 2012*). Required genome index was built with corresponding GRCm38 fasta file retrieved from Ensembl (http://ftp.ensembl.org/pub/release-102/). Multimapped reads and those exhibiting MAPQ <20 were removed. Randomly placed multi-mappers were removed from the mapped reads. For all cases, duplicated reads were marked with SAMtools (v1.15) (*Danecek et al., 2021*). *Drosophila melanogaster* spike-ins, present in H3K27me3, were discarded. NSC and RSC quality metrics were computed with PhantomPeakQualTools (v1.2) (*Landt et al., 2012*). ENCODE blacklisted regions (mm10 v2) were removed prior to peak calling. BigWig files were individually generated using deep-Tools (v3.5.1) bamCoverage with -ignoreDuplicates -binSize 10 -smoothLength 30 – effectiveGenome-Size 2308125349 -normalize Using RPGC and -extendReads *Fragment_Length* options (*Ramírez et al., 2016*). The effective genome size was particularized to a read length of 50 bp and directly retrieved from deepTools web site (https://deeptools.readthedocs.io/en/develop/content/feature/effectiveGenomeSize.html). *Fragment_Length* was retrieved from PhantomPeakQualTools results. For all histone marks except for H3K4me3, peak calling was conducted by means of epic2 (v0.0.52) with -effective- genome-fraction 0.8452 -fragment-size *Fragment_Length* options and chromosome sizes only referring to canonical chromosomes (*Stovner and Sætrom, 2019*). For H3K4me3, MACS2 (v2.2.7.1) was used to identify peaks with -nomodel -extsize *Fragment_Length* -g 2308125349 options (*Zhang et al., 2008*). The corresponding input sample was used in all peak calling computations. For the histone marks, peaks were called with adjusted p-values (FDR)<0.05.

Peak annotation was performed with ChIPseeker R (v4.2.1) package (v1.34.1) considering a TSS region range between 5000 bp upstream and 100 bp downstream (*Yu et al., 2015*). Required TxDb object was generated from the Ensembl GRCm38 gtf file retrieved (http://ftp.ensembl.org/pub/

release-102/). A consensus peakset was obtained from the three biological replicates per histone mark and condition (IκBα-WT or IκBα-KO) by identifying overlapping peaks in at least two out of the three replicates. Differential binding analysis (DBA) was conducted with DiffBind (v3.8.4) per histone mark (*Stark and Brown, 2011*). Default parameters were used except for the summits parameter which was set to 500 bp for all histone marks except for H3K4me3 (150 bp) in order to consider, for testing, intervals of 1000 bp or 300 bp, respectively. Summit values were selected to have interval widths between the minimum and first quartile peak width values for each histone mark. EdgeR was the statistical method used for all the three analyses. Differentially bound regions (DBRs) between IκBα-KO and IκBα-WT samples were called with adjusted p-values (FDR)<0.05. Identified DBRs with any annotated gene were plotted with EnrichedHeatmap R package (v1.18.1) (*Gu et al., 2018*). To summarize replicates per condition, normalizeToMatrix function was used based on corresponding BigWig files in w0 mean mode in 50 bp windows. Tracks visualization were obtained by means of Integrative Genomics Browser (IGV) tool (*Robinson et al., 2011*). List of DBA regions are listed in *Supplementary file 2*.

## Enhancers activity identification

Consensus peaksets derived from the three H3K27ac and H3K4me1 IκBα-WT mESCs replicates were used to identify poised and active enhancers in an IκBα-WT scenario. Putative poised enhancers were defined as those regions with H3K4me1 but no H3K27ac peaks. Putative active enhancers were defined as those regions with H3K27ac and presence/absence H3K4me1 peaks. To assess any differential enhancers activity in IκBα-KO vs IκBα-WT mESCs cultured in Serum/LIF, differentially bound regions and consensus peaksets (IκBα-KO) identified in H3K4me1 and H3K27ac were required. Putative gained or lost poised/active enhancers in IκBα $^{-/-}$ were obtained with the following criteria: (i) for gained/lost poised enhancers in IκBα $^{-/-}$: a differential H3K4me1 increase/decrease and absence of H3K27ac peaks in IκBα $^{-/-}$ consensus peakset, (ii) for gained/lost active enhancers (without H3K4me1) in IκBα-KO: a differential H3K27ac increase/decrease and absence of H3K4me1 in IκBα-KO consensus peakset and (iii) for gained/lost active enhancers (with H3K4me1) in IκBα-KO: a differential H3K27ac increase/decrease and presence of H3K4me1 in IκBα $^{-/-}$ consensus peakset. List of differential enhancers are listed in *Supplementary file 2*. A maximum gap of 1000 bp was allowed for checking the overlap between two different histone marks. For this purpose, the 'subsetByOverlaps' function from the IRanges R package was used (v.2.34.1).

## RNA-seq experiments

Total RNA from three independent clones from mESCs, 48 hr EBs and 96 hr EBs was isolated using the RNeasy Plus Mini Kit (Qiagen; Cat #74136) following the manufacturer's instructions. Amount of RNA was quantified with Nanodrop (Thermo Fisher; Cat #ND2000CLAPTOP), and RNA integrity was addressed by agarose gel and Agilent Bioanalyzer (Agilent Technologies; Cat #G2939BA). Libraries sequenced using Illumina HiSeq 2500 (Illumina, Inc) (125 bp paired-end reads). Samples sequencing depth ranged between 35 M and 52 M reads (average 41 M reads) per sample.

## RNA-seq data analysis

Quality control was performed on raw data with the FASTQC tool (v0.11.9). Raw reads were trimmed to remove adapter presence with Trimgalore (v0.6.6) (*Ewels and Afyounian, 2012*). Default parameters were used except for a minimum quality of 15 (Phred score) and an adapter removal stringency of 3 bp overlap. Trimmed reads were aligned to reference the genome with STAR aligner tool (v2.7.8). STAR was executed with default parameters except for the number of allowed mismatches which was set to 1. Required genome index was built with corresponding GRCm38 gtf and fasta files retrieved from Ensembl (http://ftp.ensembl.org/pub/release-102/). Obtained BAM files with uniquely mapped reads were considered for further analysis. Raw gene expression was quantified using featureCounts tool from subRead software (v2.0.1) with exon as feature (*Liao et al., 2014*). The raw counts matrix was imported into the R Statistical Software environment (v4.2.1) for downstream analysis. Raw expression matrix included 55,487 genes per 18 samples in total. Experimental design considered three timepoints: mouse ESCs, Ebs at 48 hr, and EBs at 96 hr. Each time point included six samples distributed in two conditions: 3 IκBα-WT and 3 IκBα-KO. Prior to statistical analysis, those genes with less than 10 raw counts across the six samples under test were removed. After pre-filtering, 21,843 genes (mESCs), 21,424 genes (EBs 48 hr) or 21,664 genes (EBs 96 hr) were available for testing.

For visualization purposes, counts were normalized by the variance-stabilizing transformation method as implemented in DESeq2 R package (*Love et al., 2014*) (v1.38.3). Differential expression analysis (DEA) was conducted with DESeq2. Each time point was independently analyzed. Fitted statistical model included sample conditions as covariable with IκBα-WT as the reference. Obtained log2 fold change values were shrunken with apeglm shrinkage estimator R package (v1.20.0) (*Zhu et al., 2019*). Raw p- values were adjusted for multiple testing using the Benjamini-Hochberg False Discovery Rate (FDR) (*Benjamini and Hochberg, 1995*). Differentially Expressed Genes (DEGs) between IκBα-KO and IκBα-WT samples were called with adjusted p-values (FDR)<0.05 and absolute shrunken log2 Fold change >1. Data visualization was performed with the ggplot2 (v3.4.1). List of DEGs are found in *Supplementary file 3*.

## Genome-wide DNA methylation samples preparation

DNA from frozen mESCs pellets was extracted using DNeasy Blood and Tissue Kit (Qiagen GmbH, Hilden, Germany). Purified genomic DNA was quantified with Qubit (Invitrogen, Carlsbad, CA, USA) according to the manufacturer's instructions. Infinium Mouse Methylation BeadChip (Illumina, Inc, San Diego, CA, USA) arrays were used to profile DNA methylation genome-wide. This platform allows over 285,000 methylation sites per sample to be interrogated at single-nucleotide resolution. The samples were bisulfite converted using EZ DNA Methylation-Gold Kit (Zymo Research, CA, USA) and were hybridized in the array following the manufacturer's instructions.

## DNA methylation data analysis

The DNA methylation profile of the studied samples was assessed using the Infinium Mouse Methylation BeadChip Array (~285,000 methylation sites) as previously described (*Garcia-Prieto et al., 2022*). Briefly, raw signal intensities were obtained with GenomeStudio Software 2011.1 (Illumina) and DNA methylation beta values were computed from raw IDAT files using GenomeStudio default normalization with control probes and background subtraction. Quality control steps to remove erratic probe signals were performed within the R statistical environment (v4.0.3). We removed probes with detection p-value >0.01, genotyping probes, and manufacturing flagged (MFG) probes described in the Illumina manifest file (https://support.illumina.com/downloads/infinium-mouse-methylation-manifest-file.html). The differentially methylated probes (DMPs) between IκBα-WT and IκBα-KO samples were computed separately for each time point (mESCs and EBs at 96 hr) by deriving a linear model with the limma R package (v3.46.0). Each condition included 3 samples per time point. DMPs with adjusted p-value (FDR)<0.05 and absolute mean methylation beta value difference between conditions >0.3 were considered significant. DNA methylation analysis was performed using the mm10 mouse genome reference build and the complete annotation was downloaded from the annotated manifest file (http://zwdzwd.github.io/InfiniumAnnotation#mouse; *Zhou et al., 2022*). List of differentially DNA methylated regions are listed in *Supplementary file 4*.

## Functional analysis

Overrepresentation analysis was applied over lists of selected genes derived from RNA-seq data (DEGs) or from ChIP-seq data (differentially bound regions). The Gene Ontology (Biological Process ontology, GO BP terms), KEGG PATHWAY and WikiPathways databases for *Mus musculus* (*Ashburner et al., 2000*; *Kanehisa and Goto, 2000*; *Martens et al., 2021*) were interrogated by means of cluster-Profiler R package (v4.6.2) (*Wu et al., 2021*). Corresponding Entrez identifiers were used. Benjamini-Hochberg procedure was used to obtain adjusted p-values. Obtained GO BP terms were simplified using the simplify function from clusterProfiler with default parameters. Overrepresented terms or processes were called with adjusted p-values (FDR)<0.05.

Gene Set Enrichment Analysis (GSEA) was performed for mESCs against naïve and ground state pluripotency signatures defined by other authors (*Ghimire et al., 2018*). For this purpose, the complete list of genes from mESCs samples (21,843 genes) was ranked based on the shrunken log2 Fold Change obtained from DEA. GSEA was conducted through the fgseaMultilevel function from fgsea R package (v1.24.0) (*Korotkevich et al., 2021*) with default parameters except for maxSize = 600 and eps = 0. Enrichment plots were generated with the same package.

Additionally, the testEnrichment function from SeSAMe R package (v1.14.2) (*Zhou et al., 2018*) was used for conducting functional analysis with default parameters over the list of DMPs. Probe

design, transcription factor binding site, and histone modifications consensus database sets, included in the same package, were interrogated.

Gene Set Variation Analysis (GSVA) R package (v1.46.0) with default parameters were used to obtain Z-score values for genes annotated to endoderm (GO:0001706), mesoderm (GO:0001707), and ectoderm (GO:0001705) GO BP terms (*Hänzelmann et al., 2013*).

## Statistical analysis

Statistical analysis was performed with GraphPad Prism v.8.0.1. (GraphPad Software, Inc).

Unless specified, the comparison between two groups was performed with unpaired two-sided t-test. A p-value <0.05 was considered significant.

## Acknowledgements

We would like to acknowledge all members of the Bigas/Espinosa labs for helpful discussions. We are grateful for technical support to CRG/UPF Flow Cytometry Unit, CRG Genomics and Advanced Light Microscopy Units. This work has been supported by Spanish Ministry of Science and Innovation (PID2019-104695RB-I00, PID2022-137945OB-I00, PLEC2021-007518, PDC2021-120817-I00), Generalitat de Catalunya (2021SGR 39) and Departament de Salut (SLT002/16/00299) to AB; the Spanish Ministry of Science and Innovation (PID2019- 108322 GB-100 and PID2022-142679NB-I00) to LDC; the Spanish Ministry of Science and Innovation (PID2021-123383NB-I00.) and the Agencia de Gestió d'Ajuts Universitaris i de Recerca (2021 SGR 01222) to BP. LGP has been a recipient of FI AGAUR fellowship (2019 FI- B 00151/2020 FI_B1 00130) from Generalitat de Catalunya. D.A-V was funded by the FIS fellowship (FI20/00130) from Instituto Carlos III. MB received funding from the Ramón Areces Foundation. MM is a recipient of a grant from the Instituto Carlos III, grant number CA22/00011 (co-funded by the European Social Fund Plus, ESF+, and by the European Union).

## Additional information

### Funding

| Funder | Grant reference number | Author |
| --- | --- | --- |
| Ministerio de Ciencia e Innovación | PID2019-104695RB-I00 | Anna Bigas |
| Ministerio de Ciencia e Innovación | PID2022-137945OB-I00 | Anna Bigas |
| Ministerio de Ciencia e Innovación | PLEC2021-007518 | Anna Bigas |
| Ministerio de Ciencia e Innovación | PDC2021-120817-I00 | Anna Bigas |
| Generalitat de Catalunya | 2021SGR 39 | Anna Bigas |
| Departament de Salut, Generalitat de Catalunya | SLT002/16/00299 | Anna Bigas |
| Ministerio de Ciencia e Innovación | PID2019-108322GB-100 | Luciano Di Croce |
| Ministerio de Ciencia e Innovación | PID2022-142679NB-I00 | Luciano Di Croce |
| Ministerio de Ciencia e Innovación | PID2021-123383NB-I00 | Bernhard Payer |
| Agència de Gestió d'Ajuts Universitaris i de Recerca | 2021 SGR 01222 | Bernhard Payer |
| Agència de Gestió d'Ajuts Universitaris i de Recerca | 2019 FI- B 00151 | Luis G Palma |

| Funder | Grant reference number | Author |
|---|---|---|
| Agència de Gestió d'Ajuts Universitaris i de Recerca | 2020 FI_B1 00130 | Luis G Palma |
| Instituto de Salud Carlos III | FI20/00130 | Daniel Alvarez-Villanueva |
| Instituto de Salud Carlos III | CA22/00011 | Maria Maqueda |
| Ramon Areces Foundation | | Mercedes Barrero |

The funders had no role in study design, data collection and interpretation, or the decision to submit the work for publication.

## Author contributions

Luis G Palma, Conceptualization, Formal analysis, Validation, Investigation, Methodology, Writing – original draft, Writing – review and editing; Daniel Alvarez-Villanueva, Conceptualization, Investigation, Methodology, Writing – original draft, Writing – review and editing; Maria Maqueda, Conceptualization, Resources, Data curation, Software, Formal analysis, Writing – original draft, Writing – review and editing; Mercedes Barrero, Conceptualization, Validation, Investigation, Methodology; Arnau Iglesias, Validation, Investigation; Joan Bertran, Conceptualization, Methodology; Damiana Alvarez, Carlos A Garcia-Prieto, Cecilia Ballare, Investigation, Methodology; Virginia Rodriguez-Cortez, Clara Bueno, Investigation; August Vidal, Methodology; Alberto Villanueva, Luciano Di Croce, Bernhard Payer, Manel Esteller, Conceptualization, Investigation; Pablo Menendez, Conceptualization, Investigation, Methodology; Gregoire Stik, Conceptualization; Lluis Espinosa, Conceptualization, Supervision, Funding acquisition, Investigation, Methodology, Project administration; Anna Bigas, Conceptualization, Resources, Supervision, Funding acquisition, Visualization, Writing – original draft, Project administration, Writing – review and editing

## Author ORCIDs

Anna Bigas (iD) https://orcid.org/0000-0003-4801-6899

## Ethics

2-4 months old NOD.Cg-Prkdcscid Il2rgtm1Wjl/SzJ mice (The Jackson Laboratories, Strain#005557) were used for experiments, regardless of their gender, and similar numbers offemale and male mice were used in experiments. Animals were kept under pathological-free conditions, and all animal-related work was approved by the Animal CareCommittee of the Barcelona Biomedical Research Park (PRBB) and Catalan Government(Generalitat de Catalunya), with the license number 9309. Animal care and use isapproved by the PRBB-Ethics committee and accredited by AAALAC International,following European (2010/63/UE) and Spanish (RD 53/2013) legislation.

Reviewer #1 (Public review): https://doi.org/10.7554/eLife.102784.3.sa1
Reviewer #2 (Public review): https://doi.org/10.7554/eLife.102784.3.sa2
Author response https://doi.org/10.7554/eLife.102784.3.sa3

# Additional files

## Supplementary files

MDAR checklist

Supplementary file 1. Oligonucleotides for PCR primers-sgRNAs.

Supplementary file 2. ChiPseq results for Histone marks.

Supplementary file 3. Table with RNAseq analysis.

Supplementary file 4. Table with methylation array results.

## Data availability

All sequencing data is deposited at GEO under a SuperSeries with the accession number GSE239565. Individual SubSeries can be found at: GSE239563 (RNA-seq data), GSE239564 (ChIP-seq data),

and GSE239562 (Methylation array data). Scripts used to process the generated bulk RNA-seq and ChIP-seq data are available in GitHub (copy archived at *Maqueda, 2025*).

The following dataset was generated:

| Author(s) | Year | Dataset title | Dataset URL | Database and Identifier |
|---|---|---|---|---|
| Bigas A | 2025 | Chromatin activity of IκBα mediates the exit from naïve pluripotency | https://www.ncbi.nlm.nih.gov/geo/query/acc.cgi?acc=GSE239565 | NCBI Gene Expression Omnibus, GSE239565 |

The following previously published datasets were used:

| Author(s) | Year | Dataset title | Dataset URL | Database and Identifier |
|---|---|---|---|---|
| Stunnenberg HG | 2020 | The translational landscape of ground state pluripotency | https://www.ncbi.nlm.nih.gov/geo/query/acc.cgi?acc=GSE133794 | NCBI Gene Expression Omnibus, GSE133794 |
| Meissner A | 2014 | Ground state conditions induce rapid reorganization of core pluripotency factor binding that precede global epigenetic reprogramming | https://www.ncbi.nlm.nih.gov/geo/query/acc.cgi?acc=GSE56312 | NCBI Gene Expression Omnibus, GSE56312 |

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
