## [Editor Report · eLife Assessment]

This **important** study describes a non-canonical role for IκBα in regulating mouse embryonic stem cell pluripotency and differentiation, independent of the classical NF-κB pathway. The conclusions are **convincingly** supported through orthogonal approaches and separation of function mutants. The findings add new insight into pluripotency regulation in mouse cells.

---

## [Referee Report · Reviewer #1 (Public review)]

Summary:

This study probes the role of the NF-κB inhibitor IκBa in the regulation of pluripotency in mouse embyronic stem cells (mESCs). It follows from previous work that identified a chromatin-specific role for IκBa in the regulation of tissue stem cell differentiation. The work presented here shows that a fraction of IκBa specifically associates with chromatin in pluripotent stem cells. Using three Nfkbia-knockout lines, the authors show that IκBa ablation impairs the exit from pluripotency, with embryonic bodies (an in vitro model of mESC multi-lineage differentiation) still expressing high levels of pluripotency markers after sustained exposure to differentiation signals. The maintenance of aberrant pluripotency gene expression under differentiation conditions is accompanied by pluripotency-associated epigenetic profiles of DNA methylation and histone marks. Using elegant separation of function mutants identified in a separate study, the authors generate versions of IκBa that are either impaired in histone/chromatin binding or NF-κB binding. They show that the provision of the WT IκBa, or the NF-κB-binding mutant can rescue the changes in gene expression driven by loss of IκBa, but the chromatin-binding mutant can not. Thus the study identifies a chromatin-specific, NF-κB-independent role of IκBa as a regulator of exit from pluripotency.

Strengths:

The strengths of the manuscript lie in:

(a) the use of several orthogonal assays to support the conclusions on the effects of exit from pluripotency;

(b) the use of three independent clonal Nfkbia-KO mESC lines (lacking IκBa), which increase confidence in the conclusions; and

(c) the use of separation of function mutants to determine the relative contributions of the chromatin-associated and NF-κB-associated IκBa, which would otherwise be very difficult to unpick.

Weaknesses:

No notable weaknesses remain in this revised version.

---

## [Referee Report · Reviewer #2 (Public review)]

Summary:

This manuscript investigates the role of IκBα in regulating mouse embryonic stem cell (ESC) pluripotency and differentiation. The authors demonstrate that IκBα knockout impairs the exit from the naïve pluripotent state during embryoid body differentiation. Through mechanistic studies using various mutants, they show that IκBα regulates ESC differentiation through chromatin-related functions, independent of the canonical NF-κB pathway.

Strengths:

The authors nicely investigate the role of IκBα in pluripotency exit, using embryoid body formation and complementing the phenotypic analysis with a number of genome-wide approaches, including transcriptomic, histone marks deposition, and DNA methylation analyses. Moreover, they generate a first-of-its-kind mutant set that allows them to uncouple IκBα's function in chromatin regulation versus its NF-κB-related functions. This work contributes to our understanding of cellular plasticity and development, potentially interesting a broad audience including developmental biologists, chromatin biology researchers, and cell signaling experts.

Weaknesses:

Future experiments will likely help establish a more direct mechanistic link between IκBα activity and the chromatin remodeling events observed in pluripotent cells.

---

## [Author Response]

The following is the authors’ response to the original reviews

**Public Reviews:**

**Reviewer #1 (Public review):**
Summary:This study probes the role of the NF-κB inhibitor IκBa in the regulation of pluripotency in mouse embyronic stem cells (mESCs). It follows from previous work that identified a chromatin-specific role for IκBa in the regulation of tissue stem cell differentiation. The work presented here shows that a fraction of IκBa specifically associates with chromatin in pluripotent stem cells. Using three Nfkbia-knockout lines, the authors show that IκBa ablation impairs the exit from pluripotency, with embryonic bodies (an in vitro model of mESC multi-lineage differentiation) still expressing high levels of pluripotency markers after sustained exposure to differentiation signals. The maintenance of aberrant pluripotency gene expression under differentiation conditions is accompanied by pluripotency-associated epigenetic profiles of DNA methylation and histone marks. Using elegant separation of function mutants identified in a separate study, the authors generate versions of IκBa that are either impaired in histone/chromatin binding or NF-κB binding. They show that the provision of the WT IκBa, or the NF-κB-binding mutant can rescue the changes in gene expression driven by loss of IκBa, but the chromatin-binding mutant can not. Thus the study identifies a chromatin-specific, NF-κB-independent role of IκBa as a regulator of exit from pluripotency.Strengths:The strengths of the manuscript lie in: (a) the use of several orthogonal assays to support the conclusions on the effects of exit from pluripotency; (b) the use of three independent clonal Nfkbia-KO mESC lines (lacking IκBa), which increase confidence in the conclusions; and (c) the use of separation of function mutants to determine the relative contributions of the chromatin-associated and NF-κB-associated IκBa, which would otherwise be very difficult to unpick.Weaknesses:In this reviewer's view, the term "differentiation" is used inappropriately in this manuscript. The data showing aberrant expression of pluripotency markers during embryoid body formation are supported by several lines of evidence and are convincing. However, the authors call the phenotype of Nfkbia-KO cells a "differentiation impairment" while the data on differentiation markers are not shown (beyond the fact that H3K4me1, marking poised enhancers, is reduced in genes underlying GO processes associated with differentiation and organ development). Data on differentiation marker expression from the transcriptomic and embryoid body immunofluorescent experiments, for example, should be at hand without the need to conduct many more experiments and would help to support the conclusions of the study or make them more specific. The lack of probing the differentiation versus pluripotency genes may be a missed opportunity in gaining in-depth understanding of the phenotype associated with loss of the chromatin-associated function of IκBa.
**Reviewer #2 (Public review):**
Summary:This manuscript investigates the role of IκBα in regulating mouse embryonic stem cell (ESC) pluripotency and differentiation. The authors demonstrate that IκBα knockout impairs the exit from the naïve pluripotent state during embryoid body differentiation. Through mechanistic studies using various mutants, they show that IκBα regulates ESC differentiation through chromatin-related functions, independent of the canonical NFκB pathway.Strengths:The authors nicely investigate the role of IκBα in pluripotency exit, using embryoid body formation and complementing the phenotypic analysis with a number of genome-wide approaches, including transcriptomic, histone marks deposition, and DNA methylation analyses. Moreover, they generate a first-of-its-kind mutant set that allows them to uncouple IκBα's function in chromatin regulation versus its NF-κB-related functions. This work contributes to our understanding of cellular plasticity and development, potentially interesting a broad audience including developmental biologists, chromatin biology researchers, and cell signaling experts.Weaknesses:- The study's main limitation is the lack of crucial controls using bona fide naïve cells across key experiments, including DNA methylation analysis, gene expression profiling in embryoid bodies, and histone mark deposition. This omission makes it difficult to evaluate whether the observed changes in IκBα-KO cells truly reflect naïve pluripotency characteristics.- Several conclusions in the manuscript require a more measured interpretation. The authors should revise their statements regarding the strength of the pluripotency exit block, the extent of hypomethylation, and the global nature of chromatin changes. - From a methodological perspective, the manuscript would benefit from additional orthogonal approaches to strengthen the knockout findings, which may be influenced by clonal expansion of ES cells.

Overall, this study makes an important contribution to the field. However, the concerns raised regarding controls, data interpretation, and methodology should be addressed to strengthen the manuscript and support the authors' conclusions.

**Recommendations for the authors:**

**Reviewer #1 (Recommendations for the authors):**
I have the following comments and suggestions for the authors to consider:(1) Fig, 1D: the number of replicates for this experiment is not mentioned. It would be good to see if the apparent accumulation of IκBa on chromatin of S/L cells is reproducible. If it is, does the accumulation of IκBa "prime" chromatin for differentiation?

We apologize for missing this information in the figure legend. We have repeated the experiment two independent times, and confirmed the localization of IκBα in the chromatin fraction of mESCs cultured in Serum/LIF (S/L). We have included the information in the figure legend.

Regarding the second question, we do believe that the presence of IκBα primes mESCs to exit from differentiation. Previous data from the lab (Mulero et al Cancer Cell 2012; Marruecos et al EMBO Reports 2020) demonstrated that IκBα regulates important developmental genes (Hox genes and differentiation-related genes), which become dysregulated upon IκBα depletion. Based on those previous results, together with our results that demonstrated that lack of IκBα hyperactivates the pluripotency network, we conclude that IκBα is a crucial element to attenuate pluripotency programs, allowing a successful exit from naïve pluripotency and differentiation.

(2) Fig. 1E: From what is shown, Rela doesn't agree (i.e. no enrichment in EpiSCs in the Atlasi data). Are the culture conditions in Atlasi 2020 the same as in this paper (base medium etc.)? Also, why not label all genes/proteins that are shown in 1C?

Differences observed between our data and the in-silico data might be due to differences in culture conditions used in Atlasi and colleagues. In particular, Atlasi et al. cultured the mESCs in 2i/LIF for 2 consecutive months, whereas we induced ground state of naïve pluripotency (2i/LIF) for only 96h. In the case of EpiSC differentiation, similar protocols are used in both our work and in Atlasi et al. Nevertheless, despite existing differences, in both studies IκBα is enriched in the ground state of naive pluripotency.

The reason why some proteins that are missing in Figure 1E but appearing in Figure 1C is because they are not detected in the mass spectrometry experiment.

(3) Fig. 1F: The word "clustering" here is misleading. While Nfkbia shows similar dynamics as pluripotency genes, clustering should not be used unless clusters of genes are shown in the same heatmap (and the transcripts naturally cluster together). The figure would be even more informative if all the genes from the 4 different categories were presented on the same heatmap.

As suggested by the reviewer, we have generated a heatmap where the genes from the different four categories (Figure 1F) are displayed and clustered together:

**Author response image 1. sa3fig1:** Heatmap including all the genes from Figure 1F of the manuscript and clustering is simultaneously conducted over the four categories.

As shown in previous heatmap, we can confirm that most of the Nf-kB genes (except for Nfkbia and Nfkbid) clustered together with differentiation markers.

Nonetheless, to be more conservative with original Figure 1F and for clarity upon gene categories, we have updated the figure with a combined heatmap, sliced by gene categories. In this updated version, we can observe how IkBα gene, though classified by the biological process where it classically belongs (NF-kB pathway), is higher at pluripotency, whereas it decreases upon differentiation induction, similarly as most of the pluripotency genes.

We have also changed the text accordingly and have added the following sentences in the main text (lines 121-125): “The expression pattern of *Nfkbia* was similar to the pluripotency genes whereas most of the NF-κB genes were upregulated upon differentiation, showing an analogous expression dynamics as developmental genes, as previously described”.

(4) This reviewer felt that the statement "Notably, several polycomb elements were highly expressed in mESCs, consistent with the possibility that chromatin-bound IκBα modulates PRC2 activity in the pluripotent state" (p.5, lines 125-127) is premature here. While similar expression dynamics may be consistent with a linked function, they in no way suggest this. This can be more accurately stated to point out that Nfkbia shows similar expression dynamics in pluripotency and differentiation as Polycomb component genes.

We agree that the statement is premature and we have changed it by: “Previous reports have demonstrated that chromatin-bound IκBα modulates PRC2 activity in different adult stem cell models [27]. Interestingly, we observed that most of the Polycomb target genes follow a similar expression pattern of *Nfkbia* and pluripotency, with higher expression in mESCs (Figure 1F).” (lines 125-128 in the manucript).

(5) Top of p. 6: the results are mis-attributed to Fig. 1, it should be Fig. 2.

We thank the reviewer for this observation. We have corrected it in the main text.

(6) Fig. 1B and Fig. 5I: the images of the AP stains are very difficult to see, better resolution images should be used.

We have increased both the resolution and the size of the AP colonies.

(7) Line 142 (p.6): Fig. S1B should be S1C. In general the manuscript would benefit from review of the order and labeling of the figure panels as there are a number of inconsistencies.

We have better organized the figures in the new version of the manuscript. In particular, we have reorganized the Figure S1 to have a more logical order. We have done the same for the Figure 2 and Figure 5 and they are updated in the new version of the reviewed manuscript.

(8) The authors call the phenotype of Nfkbia-KO cells a "differentiation impairment". Do the EBs shown in Fig. 2 also express differentiation markers? Do they fail to up-regulate those markers or just fail to down-regulate pluripotency markers? At the transcriptomic level the Nfkbia-KO cells still change significantly upon provision of differentiation signals (Fig. 2C), what types of gene processes underlie the differences between WT and KO cells and which processes are common? Also, based on this figure, the phenotype looks to be more of a delay than a failure in differentiation, as the cells still follow the same trajectory but lag behind the WT cells. It is difficult to discern whether this is the case based on Fig. 2E-G as we don't see the later time point (up to Day 9).

In general, with the data presented in Fig. 2C and Fig. S1, the authors show that many of the hallmarks of exit from pluripotency are impaired in Nfkbia-KO cells, as well as the general "transcriptional status" of the cells, but they don't show differentiation markers (which would be necessary to conclude an impairment in differentiation). The data should be readily available in the datasets that are in the manuscript already and it will be informative to extract and present them. The data are not currently publicly accessible (unavailable until July 2025) so it was not possible to mine them.

We appreciate the observation, and we have included more data to confirm that the IκBα-KO cells show a differentiation impairment. In the first version of the manuscript, differentiation markers are displayed from Figures 2E-G, where genes from the three germ layers (ectoderm, mesoderm and endoderm) are not activated in IκBα-KO EBs at 48h and 96h. Moreover, the volcano plot displayed in Figure S1F of the first version clearly shows a downregulation of important differentiation genes such as a *T, Eomes*, *Lhx1* and *Foxa2*. We agree that 96h EBs is an early time point to talk about differentiation impairment. For that reason, we have also included the same pluripotent and differentiation genes in 216h EBs (Figures S1F-G of the newer version of the manuscript). It is clearly observed that IκBα-KO 216h EBs maintain an upregulation of pluripotency programs which negatively correlate with a lower differentiation capability. Moreover, the impairment in the differentiation with a higher expression of pluripotency markers is confirmed by the presence of high SSEA-1 expression in IκBα-KO 216h EBs (Figure S1C of the manuscript) and alkaline phosphatase (AP) staining (Figure 2C of the manuscript). Lastly, the fact that IκBα-KO teratomas contain higher proportion of OCT3/4+ cells further confirming that IκBα-KO cells cannot differentiate because of the inability to exit from pluripotency.

Finally, generated data (and deposited in GEO repository with SuperSeries id GSE239565) is already publicly available.

(9) Fig. 5A: even if there are no global changes in NF-κB target genes, could a small subset of NF-κB target genes still mediate the IκBa effects?

We have analyzed the whole NF-κB signature, and we have identified a small cluster of genes that are differentially expressed at 96h EBs between IκBα-KO and IκBα-WT (Author response image 2). Interestingly, what we observed is the opposite as expected since we see un downregulation of that subset in the IκBα-KO 96h EBs (Author response image 3). For that reason, detected changes in the NF-κB target gene expression after deletion of Nfkbia do not support an NF-κB inhibitory role for IkBa in pluripotent ESC.

**Author response image 2. sa3fig2:** Heatmap of NF-κB genes expression at the different time points of differentiation (mESCs, 48h EBs, 96h EBs). Highlighted region marks the genes that are differentially expressed between both genotypes at 96h EBs.

**Author response image 3. sa3fig3:** Violin plot of genes from the NF-κB pathway which are differentially expressed at 96h EBs.

(10) Lines 233-238, the part of the text is repeated.

We appreciate the observation and have deleted the repeated part.

(11) The data in Fig. 5D-E make it difficult to be sure whether the conclusions on the relative subcellular localisations of the different mutants are accurate, as the chromatin-binding mutant seems to be less abundant than the other mutants (judging from the Input in Fig. 5C and also from the tubulin loading controls in Fig. 5D-E). Showing the IκBa levels in total extracts would make the interpretation of these data more robust. The authors do mention that the chromatin-binding mutant IκBa protein is consistently expressed at lower levels but they do not comment on how this may affect the data interpretation - could the lack of rescue be due to lower levels of the chromatin-binding mutant IκBa relative to the wild-type IκBa? This should be addressed in the Discussion, if not tested formally by normalising the expression levels of the different forms of IκBa in the rescue experiments.

Although protein stability is different among the SOF mutants, IκBα^ΔChromatin^ is exclusively detected in the cytoplasm, with lack of detection in the chromatin compartment (Figures 5D-E of the reviewed manuscript). For this reason, we believe that the quantitative differences in protein levels of the different mutants cannot explain the subcellular localization differences and the phenotype observed.

Nonetheless, we cannot discard that differences in the protein levels between SOF mutants can affect the rescue phenotype, and we have specified so in the discussion section of the manuscript.

(12) Lines 260-261: "Induction of i-IκBαWT and i-IκBαΔNF-κB reduced the expression levels of the naive pluripotent genes Zfp42, Klf2, Sox2 and Tbx3, which were increased by i-IκBαΔChromatin (Figure 5F)." This is not an accurate statement. The expression was not reduced by the ΔChrom mutant in the same way as it was by the WT and the ΔNF-κB mutant, but it was not increased.

We have better specified the description of the results displayed in Figure 5F (lines 258-261 of the main manuscript):

“Induction of i-IκBα^WT^ and i-IκBα^ΔNF-κB^ reduced the expression levels of the naïve pluripotent genes *Zfp42*, *Klf2*, *Sox2* and *Tbx3*. On the other hand, the same genes either do not change their expression (*Zfp42*, *Sox2*, *Klf2*) or increase their levels (*Tbx3*) upon i-IκBα^ΔChromatin^ induction (Figure 5F).”

(13) In Fig. 5J the images will ideally be shown before and after Doxycycline treatment, to better support the conclusions.

We have included a new panel in Figure S4 (Figure S4E in the reviewed manuscript) where the No doxycycline control 216 EBs between the different conditions (i-IκBα^WT^, i-IκBα^ΔChrom^ and i-IκBα^ΔNF-κB^) are included.

**Reviewer #2 (Recommendations for the authors):**
- The PCA analysis in Figure 2 appears to contradict the authors' conclusions about global transcriptome changes in KO cells. Furthermore, there is a discrepancy between immunofluorescence data showing near-complete methylation loss and the methylation array analysis results.

Although there is a differentiation block in the IkBa KO EBs, this is not complete and they show some differentiation trend after 96h (Fig 2C), moreover, acquisition of differentiation genes from all three germ layers is strongly affected (Figure 2E of the reviewed manuscript) and these programs remain downregulated and pluripotency genes are still expressed in IκBα-KO EBs at later time points (216h) (Fig 2B). Altogether demonstrates that the lack of IκBα impairs differentiation and the silencing of the pluripotency network.

Discrepancies between methylation array and immunofluorescence are expected since immunofluorescence is not quantitative and the methylation array is very precise.

- The authors should revise their statements regarding the strength of the pluripotency exit block, the extent of hypomethylation, and the global nature of chromatin changes. For example, the observed chromatin changes, including H3K27ac modifications, appear relatively modest and should be described as such. - The manuscript would benefit from additional orthogonal approaches to strengthen the knockout findings, which may be influenced by clonal expansion of ES cells. Additionally, the emphasis on overlapping H3K4me3 and H3K27me3 regions should be reduced, as these represent a minor fraction of the affected regions (only 41 regions).

We have revised the text and have included it in the discussion section the following text (lines 327-331 in the reviewed manuscript):

“Although IκBα KO mESCs exhibit a transcriptional phenotype and hypomethylation state that resembles the ground state of naïve pluripotency, there are only modest changes on histone marks associated to enhancers (H3K27Ac) or gene regulation (H3K4me3 and H3K27me3). Altogether indicates that further experiments are required to fully elucidate the effect of chromatin IκBα.”

We have also included Fig S3E-S3F to show that similar differences as WT and KO in H3K4me3 and H3K27me3 are observed in a serum/LIF and 2i conditions, further supporting the fact that KO cells in Serum/LIF resemble WT cells in 2i condition.